



# Glacial inception through rapid ice area increase driven by albedo and vegetation feedbacks

Matteo Willeit[1], Reinhard Calov[1], Stefanie Talento[1], Ralf Greve[2,3], Jorjo Bernales[4], Volker Klemann[5], Meike Bagge[5], and Andrey Ganopolski[1]

[1]Potsdam Institute for Climate Impact Research (PIK), Member of the Leibniz Association, P.O. Box 601203, D-14412 Potsdam Germany
[2]Institute of Low Temperature Science, Hokkaido University, Sapporo, Japan
[3]Arctic Research Center, Hokkaido University, Sapporo, Japan
[4]Institute for Marine and Atmospheric Research Utrecht, Utrecht University, Utrecht, The Netherlands
[5]GFZ German Research Centre for Geosciences, Department of Geodesy, Potsdam, Germany

**Correspondence:** Matteo Willeit (willeit@pik-potsdam.de)

**Abstract.**

We present transient simulations of the last glacial inception using the Earth system model CLIMBER-X with dynamic vegetation, interactive ice sheets and visco-elastic solid-Earth response. The simulations are initialized at the middle of the Eemian interglacial (125 kiloyears before present, ka) and run until 100 ka, driven by prescribed changes in Earth's orbital parameters and greenhouse gas concentrations from ice core data.

CLIMBER-X simulates a rapid increase in Northern Hemisphere ice sheet area through MIS5d, with ice sheets expanding over northern North America and Scandinavia, in broad agreement with proxy reconstructions. While most of the increase in ice sheet area occurs over a relatively short period between 119 ka and 117 ka, the larger part of the increase in ice volume occurs afterwards with an almost constant ice sheet extent.

We show that the vegetation feedback plays a fundamental role in controlling the ice sheet expansion during the last glacial inception. In particular, with prescribed present-day vegetation the model simulates a global sea level drop of only ~20 m, compared with the ~35 m decrease in sea level with dynamic vegetation response. The ice sheet and carbon-cycle feedbacks play only a minor role during the ice sheet expansion phase prior to ~115 ka, but are important in limiting the deglaciation during the following phase characterized by increasing summer insolation.

The model results are sensitive to climate model biases and to the parameterisation of snow albedo, while they show only a weak dependence on changes in the ice sheet model resolution and the acceleration factor used to speed up the climate component.

Overall, our simulations confirm and refine previous results showing that climate-vegetation-cryosphere-carbon cycle feedbacks play a fundamental role in the transition from interglacial to glacial states characterising Quaternary glacial cycles.





## 1  Introduction

Modeling of glacial inceptions, i.e. the relatively fast (compared to the duration of glacial cycles) transitions of the Earth system from an interglacial (no significant ice sheets over the northern continents) to a glacial state, caused by lowering of the boreal summer insolation, is a crucial test for the Milankovitch theory of glacial cycles (Milankovitch, 1941). Milutin Milanlkovitch was the first who used mathematical modelling to test the hypothesis that the lowering of boreal summer insolation associated

with precession and obliquity cycles is sufficient to cause growth of northern ice sheets. In his "Cannon of insolation", using a simple energy-balance model and taking into account the ice-albedo feedback, Milankovitch estimated that a typical decrease in summer insolation would cause a lowering of the snow line by about 1 km and would cause a southward expansion of ice cover up to 55N.

Since the most recent glacial inception, which began between 120 and 115 ka, is relatively well covered by paleoclimate

data and occurred during one of the deepest (the 2nd lowest during the late Quaternary) boreal summer insolation minima, it is often used for testing Milankovitch theory. Until recently, complex climate models based on general circulation models (GCMs) did not include ice sheet components and thus could not simulate the evolution of ice sheets, which can be directly compared with existing paleoclimate data. So-called time-slice simulations with fixed orbital parameters and greenhouse gases (GHGs) concentrations corresponding to the insolation minimum (115 or 116 ka) have been traditionally performed instead.

In such simulations, an establishment of perennial snow cover over at least several model grid cells in northern North America and/or Eurasia was considered to be the criterium for a successful simulation of glacial inception. A number of such simulations have been performed during the past four decades (e.g. Royer et al., 1983; Rind et al., 1989; Dong and Valdes, 1995; Vavrus, 1999; Vettoretti and Peltier, 2011; Jochum et al., 2012). Similar time-slice experiments using climate model output to force a stand-alone ice sheet model have also been performed (e.g. Born et al., 2010). Overall, the results of these simulations

in terms of simulated (or not simulated) areas of perennial snow cover and their locations have been diverse. In many simulations, perennial snow cover did not appear even under such extreme orbital forcing or appeared in locations not supported by paleoclimate reconstructions. This is not surprising, since climate models used for paleoclimate studies usually have a coarse spatial resolution, which has been shown to have a significant impact on simulations of glacial inception (e.g. Vavrus et al., 2011; Birch et al., 2017). Not less important is the fact that climate models tend to have climate biases often comparable in

magnitude with the simulated climate response to orbital forcing. Some studies (e.g. Vettoretti and Peltier, 2003) demonstrated a significant influence of climate biases on the simulation of glacial inception. It was also proposed that the failures to simulate glacial inception in climate models can be related to the omission of some positive feedbacks, such as the climate-vegetation feedback (De Noblet et al., 1996). Several studies demonstrated that a weakening of the AMOC during glacial inception (e.g. Yin et al., 2021) could potentially affect ice growth in North America (Khodri et al., 2001) or Scandinavia (Lofverstrom et al.,

2022). The role of GHGs was also investigated (e.g. Vettoretti and Peltier, 2011), but since the last glacial inception began when concentrations of $CO_2$ and other GHGs were close to pre-industrial, it is clear that climate-carbon cycle feedbacks can contribute to the rate of ice sheet growth but not to the initial ice growth. So far, only a few attempts have beeen made to simulate the last glacial inception with ice sheet models asynchronously coupled to atmospheric GCMs (Gregory et al., 2012;



Herrington and Poulsen, 2012; Tabor and Poulsen, 2016), but have generally failed to grow enough ice to match sea level
reconstructions, even when using permanent coldest orbit (116 ka). Many more simulations of the last glacial inception and
even full glacial cycles have been performed with different types of simplified, geographically explicit climate-ice coupled
models. In particular, 2-D energy balance climate models were coupled with 2-D (Peltier and Marshall, 1995; Tarasov and
Peltier, 1997) and later 3-D thermomechanical ice sheet models (Tarasov and Peltier, 1999) to simulate the last glacial cycle.
More recently, different Earth system models of intermediate complexity (EMICs) were applied in transient simulations of the
last glacial inception and the entire last glacial cycle (or even several glacial cycles) (Calov et al., 2005a; Bonelli et al., 2009;
Ganopolski et al., 2010; Heinemann et al., 2014; Ganopolski and Brovkin, 2017; Bahadory et al., 2021). A set of transient
and equilibrium experiments with the EMIC CLIMBER-2 (Calov and Ganopolski, 2005) demonstrated that glacial inception
could be interpreted as a bifurcation transition in the Earth system in the phase space of orbital forcing. A large ensemble of
simulations with the LOVECLIM EMIC (Bahadory et al., 2021) explored the dependence of simulated glacial inception on
model parameters. While EMICs are much more computationally efficient than GCMs, which allows to perform many long-
term (orbital time scale) simulations, they usually have even larger climate biases than GCMs. This often causes significant
misplacement of simulated ice sheets during glacial inception compared to paleoclimate reconstructions (Calov et al., 2005a;
Bonelli et al., 2009; Bahadory et al., 2021). It was shown that the correction of temperature biases significantly improves the
agreement between simulated and reconstructed ice sheets evolution during glacial inception (Ganopolski et al., 2010).

In this study we employ the Earth System model CLIMBER-X (Willeit et al., 2022, 2023) with interactive ice sheets, visco-
elastic solid Earth response and dynamic vegetation to simulate the last glacial inception from 125 ka to 100 ka.

## 2   Model description and setup

To simulate the last glacial inception we use the Earth system model CLIMBER-X (Willeit et al., 2022, 2023). Its climate
model component includes a semi-empirical, statistical-dynamic atmosphere model, a 3D frictional-geostrophic ocean model,
a dynamic-thermodynamic sea ice model and a land surface and dynamic vegetation model, all with a horizontal resolution of
5x5 degrees.

Additionally, the two ice sheet models SICOPOLIS (Greve, 1997) and Yelmo (Robinson et al., 2020) are part of CLIMBER-
X (Willeit et al., 2022). In the present study we employ SICOPOLIS to model the Northern Hemisphere (NH) ice sheets, since
it has already been extensively applied to NH glaciation studies, particularly in the framework of the CLIMBER-2 model (e.g.
Calov et al., 2005a, 2009; Ganopolski et al., 2010; Willeit et al., 2019), the predecessor of CLIMBER-X. In CLIMBER-X
we make use of one of the latest versions of SICOPOLIS (v5.1;SICOPOLIS Authors (2019)), which is based on the shallow
ice approximation for grounded ice, the shallow shelf approximation for floating ice and hybrid shallow-ice–shelfy-stream
dynamics for ice streams (Bernales et al., 2017). The enthalpy method of Greve and Blatter (2016) is used as thermodynamics
solver. The ice sheet model is applied to the NH at a default horizontal resolution of 32 km, while Antarctica is prescribed at
its present-day state in this study. The coupling of the ice sheet model to the other CLIMBER-X components is schematically
illustrated in Fig. 1 and further details of SICOPOLIS are presented in Appendix A.





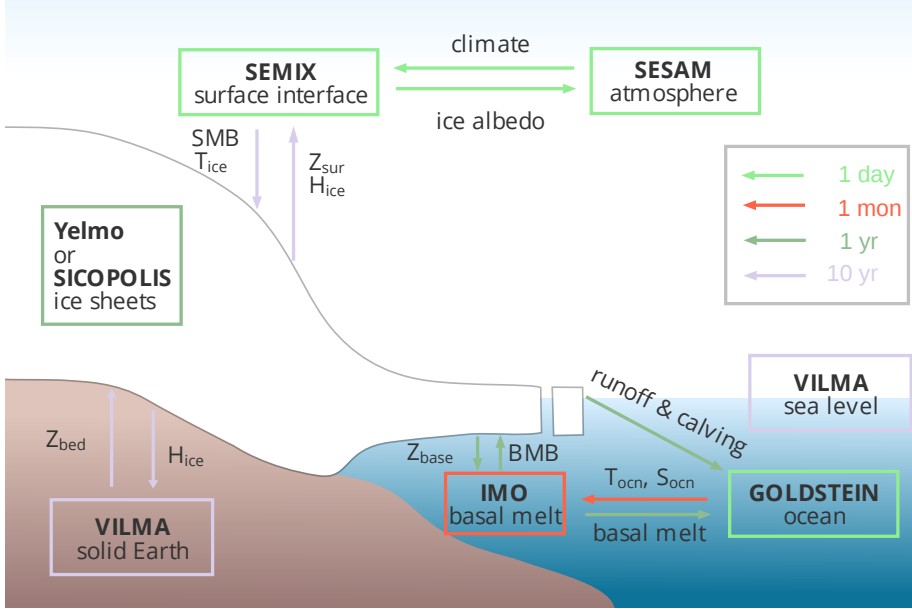

**Figure 1.** Schematic illustration of the coupling of the ice sheets to the other CLIMBER-X model components. The colors of the arrows indicate the coupling frequency, while the colors of the boxes represent the internal time step of the different model components. The abbreviations in the figure are: SMB = surface mass balance, $T_{\mathrm{ice}} = 10\,\mathrm{m}$ firn temperature, $Z_{\mathrm{sur}}$ = surface elevation, $H_{\mathrm{ice}}$ = ice thickness, $Z_{\mathrm{bed}}$ = bedrock topography, $Z_{\mathrm{base}}$ = ice sheet base elevation, BMB = basal mass balance, $T_{\mathrm{ocn}}$ = seawater temperature and $S_{\mathrm{ocn}}$ = seawater salinity.

A physically-based surface energy and mass balance scheme (SEMIX, Surface Energy and Mass balance Interface for CLIMBER-X) is used to interface the ice sheet model with the atmosphere. In the ice sheet modelling community it is still common to use the simple positive-degree-day (PDD) model (e.g. Braithwaite, 1984; Reeh, 1991) for the computation of sur-

face melt. However, this simplified approach computes melt only from near-surface air temperature, with potentially important implications on the simulated response of the surface mass balance to orbital forcing and climate change (e.g. Bauer and Ganopolski, 2017; Van De Berg et al., 2011; Bougamont et al., 2007). To address some of the limitations of the PDD scheme, so-called insolation-temperature methods (ITMs) have been developed that explicitly account for absorbed surface solar radiation (Robinson et al., 2010; van den Berg et al., 2008; Pollard, 1980; Pellicciotti et al., 2005; Krebs-Kanzow et al., 2018).

Recently, energy balance schemes or land surface models have also been applied to compute the surface mass balance of ice sheets (Kapsch et al., 2021; Lipscomb et al., 2013; Ackermann et al., 2020, e.g.). Similarly to these models, SEMIX does also explicitly resolve the surface energy balance and is largely based on the ideas in Calov et al. (2005a), but with notable modifications and improvements. The climate model fields needed by SEMIX are first bilinearly interpolated from the coarse resolution climate model grid (5x5) onto the high-resolution ice sheet model grid, where SEMIX operates. Subsequently, temperature,

humidity and radiation fields are downscaled onto the high-resolution topography. Snow albedo depends on snow grain size and the concentration of dust in snow, while bare ice sheet albedo transitions between firn-like values, clean ice values and





dirty ice values depending on the age of ice and the time it has been exposed to the atmosphere. The surface energy balance equation is then solved following a similar approach as over land and sea ice as described in Willeit and Ganopolski (2016) and Willeit et al. (2022) and the surface mass balance is finally derived accounting for snowfall, rainfall, evaporation/sublimation,
melt, refreezing and runoff. In order to compute the annual surface mass balance, SEMIX is called every 10 years over a full year with a time step of one day. As long as the forcing is slow enough a higher frequency of SEMIX calls is not needed to accurately capture the climatological evolution of the surface mass balance, because CLIMBER-X does not resolve weather and internal inter-annual variability. SEMIX is described in detail in Appendix B.

The surface mass balance of ice sheets is largely controlled by air temperature and precipitation and relatively small climate
model biases can result in large changes in simulated ice sheets (e.g. Niu et al., 2019). In accordance with Milankovitch theory, ice sheets are particularly sensitive to summer temperatures, which control snow and ice melt. Thus, even small summer temperature biases, which are common even in state-of-the-art climate models (e.g. Fan et al., 2020), can therefore preclude a realistic simulation of ice sheet growth and decay. In CLIMBER-X, present-day summer temperature biases over NH land are generally within a few degrees (Fig. 2a). However, in the area around the Hudson Bay and the Labrador Peninsula simulated
summer temperatures are up to $\sim 5\,^{\circ}\mathrm{C}$ too warm. Biases of this order of magnitude are comparable to the temperature changes induced by changing orbital configuration (Fig. 2b) and can consequently strongly affect the simulated ice sheet distribution. Therefore, similarly to Ganopolski et al. (2010), in SEMIX we implemented a temperature bias correction over northern North America that has a dipole structure as shown in Fig. B1. The bias correction is applied throughout the year and at all times as a constant offset in the computation of the surface energy balance, assuming that the bias is a persistent feature of the model
also under different boundary conditions. Furthermore, since air temperature is closely related to the downwelling longwave radiation at the surface, which affects the surface energy balance, we additionally correct the downwelling longwave radiation using a simple quadratic relation derived from ERA5 reanalysis data. For more details see Appendix B2.

As soon as an ice sheet gets in contact with the ocean its mass balance is also influenced by melting and freezing at the base of the floating ice shelves. At present, basal melt is important mostly for the Antarctic ice sheet, which is in extensive
contact with the ocean, and for the interactions of the Greenland ice streams with the fjords. However, little is known about the role played by basal melt for ice sheets in the NH during the last glacial cycle. The basal melt is strongly influenced by the ocean circulation on the continental shelf and by the turbulent plume dynamics generated at the ice-water interface by the density difference between the fresh meltwater and the surrounding saline sea-water (e.g. Jenkins, 1991). These processes can not be properly represented at the typically applied resolutions of global ocean circulation models, and several simple
parameterisations have been put forward to represent the basal melt process in ice sheet models based on large scale ambient properties of sea water and on the ice shelf base topography (Beckmann and Goosse, 2003; Holland et al., 2008; Pollard and Deconto, 2012). Simple box models have also been developed to describe the ocean circulation in the cavities below the ice shelves (Olbers and Hellmer, 2010; Reese et al., 2018; Pelle et al., 2019), but they are specifically designed to be applied to Antarctica and an extension to the NH is not straightforward. In CLIMBER-X we have therefore implemented the simple and
general basal melt parameterisations of Beckmann and Goosse (2003) and Pollard and Deconto (2012), both of which rely on the difference between the ambient ocean temperature simulated by the ocean model and the freezing point temperature of





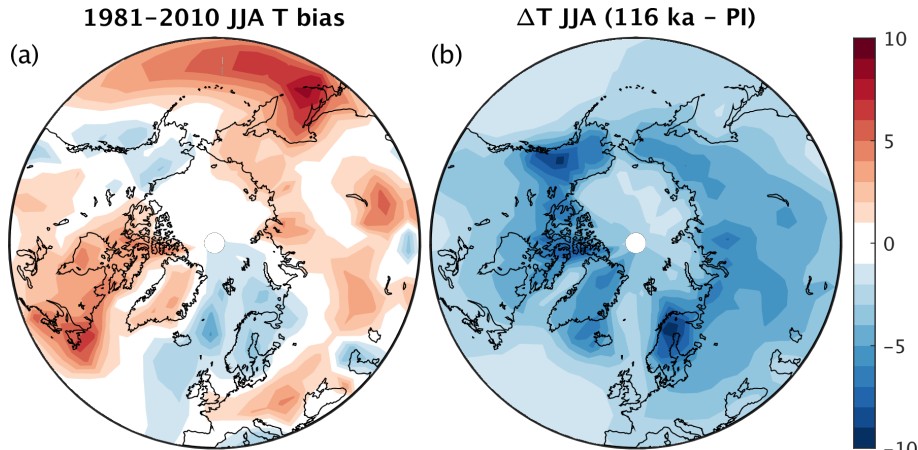

**Figure 2.** (a) CLIMBER-X summer (JJA) near-surface air temperature bias relative to ERA5 reanalysis (Hersbach et al., 2020) for the time period 1981-2010. (b) Difference in simulated summer (JJA) near-surface air temperature between 116 ka and pre-industrial resulting only from the different orbital parameters.

sea-water at the ice shelf base. The linear model of Beckmann and Goosse (2003) has been used in the simulations presented in this work. A more general 2-D plume model including also the Coriolis effect that is general enough to be applied to any ice shelf geometry has been recently developed (Lambert et al., 2022) and will be considered for a future implementation in

CLIMBER-X. See Appendix C for more details on the basal melt schemes in CLIMBER-X.

The presence of ice sheets also affects the simulated freshwater fluxes into the ocean. Freshwater produced by melt occuring at the surface of the ice sheets is routed to the ocean following the steepest surface gradient and the same runoff directions are used also to route the water originating from basal melt of grounded ice to the ocean. The freshwater flux from basal melt of floating ice and ice shelf calving is applied locally to the ocean model. Both liquid water runoff and calving fluxes are

distributed uniformly over several coastal grid cells. For calving fluxes also the latent heat needed to melt the ice is accounted for in the ocean model. CLIMBER-X does so far not include an iceberg model to explicitly simulate the fate of ice originating from ice shelf calving.

The glacial isostatic adjustment (GIA), which controls regional sea level change and surface displacements, is computed by the viscoelastic solid-Earth model VILMA (Klemann et al., 2008; Martinec et al., 2018; Bagge et al., 2021). In VILMA, mass

conservation between ice and ocean water is considered, and the loading effect of the ice and gravity-consistent redistribution of water in the ocean is solved applying the sea-level equation (Farrell and Clark, 1976) including the rotational feedback (Martinec and Hagedoorn, 2014). VILMA takes the ice sheet loading as input and computes the relative sea level (RSL) changes. The sea-level equation at the surface is solved on a Gauss-Legendre grid of 512x256 grid points (n128), which is consistent with the spectral resolution in spherical harmonics up to Legendre degree/order 170, corresponding to a wavelength

of ~120 km. In this study we use the 3D viscosity structure from Bagge et al. (2021), which is based on a 3D tomography model of the upper mantle seismic velocity structure that is converted into viscosity variations considering further constraints by



geodynamics and lithospheric structure. In CLIMBER-X, the solid Earth dynamics determines the bedrock elevation through simulated changes in relative sea level. The relative sea level computed by VILMA is added as an anomaly on top of the reference high-resolution ($\approx 10\,\mathrm{arc-minutes}$) bedrock elevation from Schaffer et al. (2016). This is done in order to preserve

the small-scale structure, which is important both for runoff routing and for surface mass balance. Surface elevation, land-sea mask and runoff routing directions are then updated accordingly. The surface elevation is then aggregated to the coarse climate model resolution and the land and ocean grid cell fractions on the climate model grid are derived. The climate model in CLIMBER-X can deal with land/sea mask changes, a feature that is rather exceptional for contemporary Earth system models (e.g. Meccia and Mikolajewicz, 2018; Riddick et al., 2018). VILMA is called every 10 years, implying that the ice sheet model

gets an update of the bedrock elevation field every 10 years and that the topography, the land-sea mask and the runoff directions are also updated every 10 years. Experiments using a higher coupling frequency of one year showed negligible differences.

## 3   Experimental design

We first performed a transient Holocene simulation from 10 ka to present-day (2000 CE) to make sure that no glacial inception is simulated at present. Since for this purpose we are mainly interested in the growth of ice sheets outside of Greenland, the

Holocene simulation was performed with a prescribed present-day Greenland ice sheet. In this simulation, the model is driven by prescribed changes in the orbital configuration and atmospheric concentrations of $CO_2$, $CH_4$ and $N_2O$ from ice core data and historical observations (Köhler et al., 2017). Note that NH summer insolation is continuosly decreasing over the Holocene, although not enough to cause a glacial inception at present (Ganopolski et al., 2016). Consistently, at the end of the transient Holocene simulation, ice cover is simulated only over parts of Baffin- and Ellsemere Island, in the south-east of Iceland, on

Svalbard, over part of Novaya Zemlya and in a small area over the Rocky Mountains (Fig. 3), in good agreement with glacier inventories (IPCC).

The transient last glacial inception simulations are started from the Eemian interglacial at 125 ka and run until 100 ka. This time interval is characterized first by a decrease in NH summer insolation, with a minimum reached at ~116 ka followed by an increase in insolation until ~105 ka (Fig. 4a). The model is driven by prescribed changes in the orbital parameters and

atmospheric concentrations of $CO_2$, $CH_4$ and $N_2O$ from ice core data (Köhler et al., 2017) (Fig. 4b). The initial conditions of the model runs correspond to the climate model in equilibrium with 125 ka boundary conditions and the Greenland ice sheet prescribed from present-day observations with a uniform ice temperature of -10 °C. With this general setup we performed different simulations to test the importance of different feedback mechanisms, namely (i) vegetation feedback, (ii) ice-sheet feedback and (iii) carbon cycle feedback, and additional experiments to explore the sensitivity of the simulated last glacial

inception to (i) the temperature bias correction in SEMIX, (ii) snow albedo parameterisation, (iii) ice sheet model resolution and (iv) climate acceleration factor. The full set of experiments is listed in Table. 1.



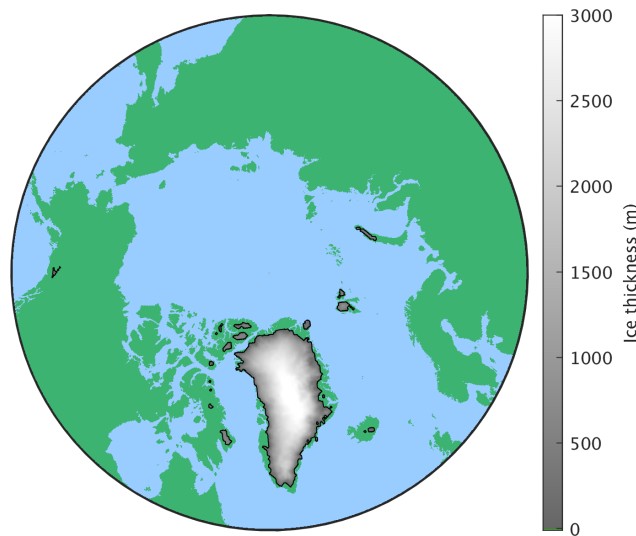

**Figure 3.** Present-day (2000 CE) ice sheet extent and thickness from the reference transient Holocene CLIMBER-X simulation. The Greenland ice sheet is prescribed in this experiment.

## 4 Modelling results

### 4.1 Reference transient simulation of the last glacial inception

In the transient simulation of the last glacial inception, until ∼120 ka only minor changes in ice sheet area (Fig. 4c) and sea level
(Fig. 4d) are produced by the model. This is followed by a rapid increase in ice sheet area by ∼10 million km² between ∼119 ka
and ∼117 ka (Fig. 4c), driven by the further decrease in NH summer insolation (Fig. 4a). Afterwards, ice area stabilizes, while
ice volume continues to grow and sea level drops to ∼-35 m by ∼110 ka. After ∼110 ka both ice area and volume decrease
again following an increase in summer insolation and despite a global cooling induced by a pronounced decrease in the
atmospheric concentration of greenhouse gases, mainly $CO_2$ (Fig. 4b). There is an overall good agreement between simulated
and reconstructed sea level in terms of timing, while the amplitude of the changes is somewhat underestimated (Fig. 4d). Part
of this discrepancy could be related to the missing contribution from Antarctica, which is prescribed at its present-day state in
our simulations.

A two-stage relation between ice sheet area and ice sheet volume in the first part of the glacial inception simulation is
clearly shown in Fig. 5. Initially, a substantial increase in ice area occurs at a nearly constant ice volume, followed by a large
increase in ice volume while the ice area remains almost constant. During the deglaciation phase no two-stage behavior in the
area–volume relation is found.

At 121 ka the simulated ice sheet extent is comparable to the present-day, but with some loss of ice in southern Greenland
(Fig. 6a). Then ice starts to rapidly expand over the Canadian Arctic archipelago and over Scandinavia (Fig. 6b). Subsequently,
starting from the Arctic islands, ice also covers the Barents and Kara Seas and the Cordilleran ice sheet is established (Fig. 6c).



**Table 1.** List of last glacial inception experiments.

| Experiment | ice model | T bias corr | T offset | vegetation | geo | GHGs | dust scaling | snow albedo offset | ice sheet resolution | acceleration |
|---|---|---|---|---|---|---|---|---|---|---|
| Ref | on | on | 0 °C | dyn | on | var | 1 | 0 | 32 km | 1 |
| NoIce | off | - | 0 °C | dyn | on | var | 1 | 0 | - | 1 |
| NoIceFixVeg | off | - | 0 °C | fix PI | on | var | 1 | 0 | - | 1 |
| FixVeg | on | on | 0 °C | fix PI | on | var | 1 | 0 | 32 km | 1 |
| FixGeo | on | on | 0 °C | dyn | off | var | 1 | 0 | 32 km | 1 |
| FixGHG | on | on | 0 °C | dyn | on | fix 125 ka | 1 | 0 | 32 km | 1 |
| NoBiasCorr | on | off | 0 °C | dyn | on | var | 1 | 0 | 32 km | 1 |
| Tm1 | on | on | -1 °C | dyn | on | var | 1 | 0 | 32 km | 1 |
| Tp1 | on | on | +1 °C | dyn | on | var | 1 | 0 | 32 km | 1 |
| Dust05 | on | on | 0 °C | dyn | on | var | 0.5 | 0 | 32 km | 1 |
| Dust2 | on | on | 0 °C | dyn | on | var | 2 | 0 | 32 km | 1 |
| $\alpha-$ | on | on | 0 °C | dyn | on | var | 1 | -0.025 | 32 km | 1 |
| $\alpha+$ | on | on | 0 °C | dyn | on | var | 1 | +0.025 | 32 km | 1 |
| Res16 | on | on | 0 °C | dyn | on | var | 1 | 0 | 16 km | 1 |
| Res64 | on | on | 0 °C | dyn | on | var | 1 | 0 | 64 km | 1 |
| Acc2 | on | on | 0 °C | dyn | on | var | 1 | 0 | 32 km | 2 |
| Acc5 | on | on | 0 °C | dyn | on | var | 1 | 0 | 32 km | 5 |
| Acc10 | on | on | 0 °C | dyn | on | var | 1 | 0 | 32 km | 10 |
| Acc20 | on | on | 0 °C | dyn | on | var | 1 | 0 | 32 km | 20 |
| Acc50 | on | on | 0 °C | dyn | on | var | 1 | 0 | 32 km | 50 |
| Acc100 | on | on | 0 °C | dyn | on | var | 1 | 0 | 32 km | 100 |

Then the ice generally grows thicker and slowly expands to reach its maximum extent at ∼115 ka (Fig. 6d). Between 115 ka and 110 ka, ice sheets mainly grow thicker with little change in ice extent (Fig. 6e). Towards the end of the simulation at ∼100 ka, about two thirds of the maximum 'glacial' ice is melted (Fig. 6f).

The simulated ice sheet cover at 110 ka compares reasonably well with the reconstructions of Dalton et al. (2022) and Batchelor et al. (2019) for marine isotope stage (MIS) 5d (Fig. 7), considering also the relatively large associated uncertainty

range (shown by the best and maximum reconstructed ice extent in Fig. 7). Model and reconstructions show a good agreement in ice sheet cover over Fennoscandia, Iceland and Greenland. Simulated ice cover is possibly underestimated over eastern North America, particularly over the Labrador region, although reconstructions are highly uncertain over this region. A Cordilleran ice sheet is formed in the model, while Alaska remains largely ice-free. Contrary to what is indicated by the reconstructions





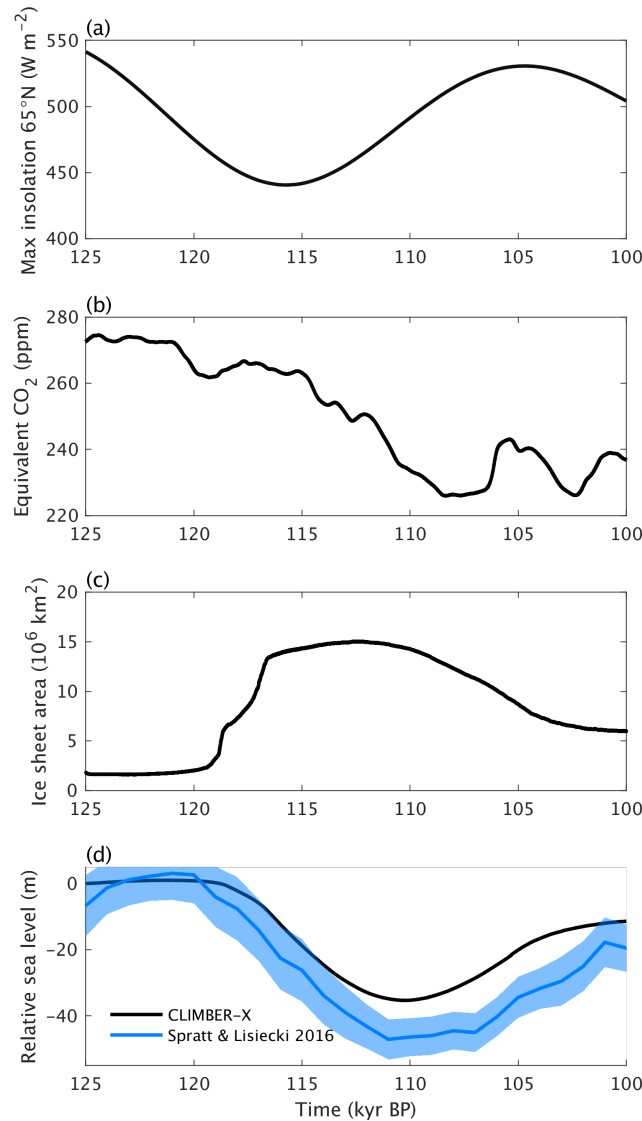

**Figure 4.** Transient last glacial inception simulation with reference model parameters. (a) Maximum summer insolation at $65\,°N$ (Laskar et al., 2004), (b) Equivalent $CO_2$ concentration for radiation (including the radiative effect of $CO_2$, $CH_4$ and $N_2O$ (Köhler et al., 2017)). (c) Simulatd ice sheet area. (d) Simulated global relative sea level change compared to reconstructions (Spratt and Lisiecki, 2016). The blue shading indicates the one standard deviation uncertainty range.

mentioned above, the Cordilleran and Laurentide ice sheets merge in the model simulation. Little ice cover is simulated by the

model over Eastern Siberia, where reconstructions suggest the presence of small ice caps covering the mountain ranges.

The relative sea level represents the geoid height change relative to the displaced Earth surface. Its signature at $110\,ka$, shown in Fig. 8, mainly represents the sum of the viscoelastic subsidence due to the loading of the ice sheet as geoid changes





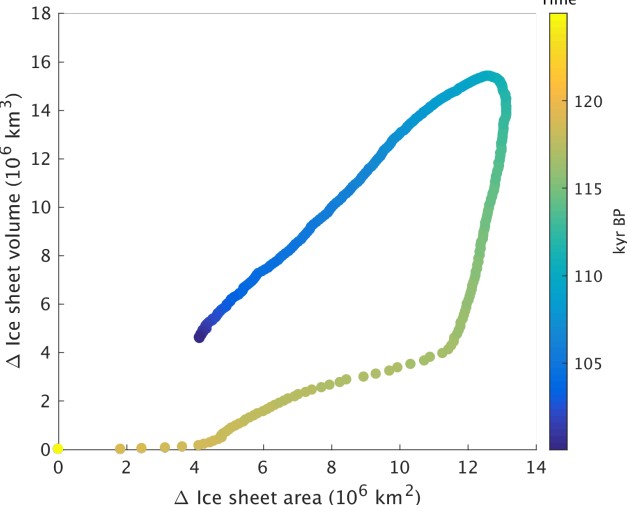

**Figure 5.** Relation between NH ice sheet area changes and ice sheet volume changes for the glacial inception simulation with the reference model version. Symbols indicate the area and volume every 50 years. The color scale shows the progression of time.

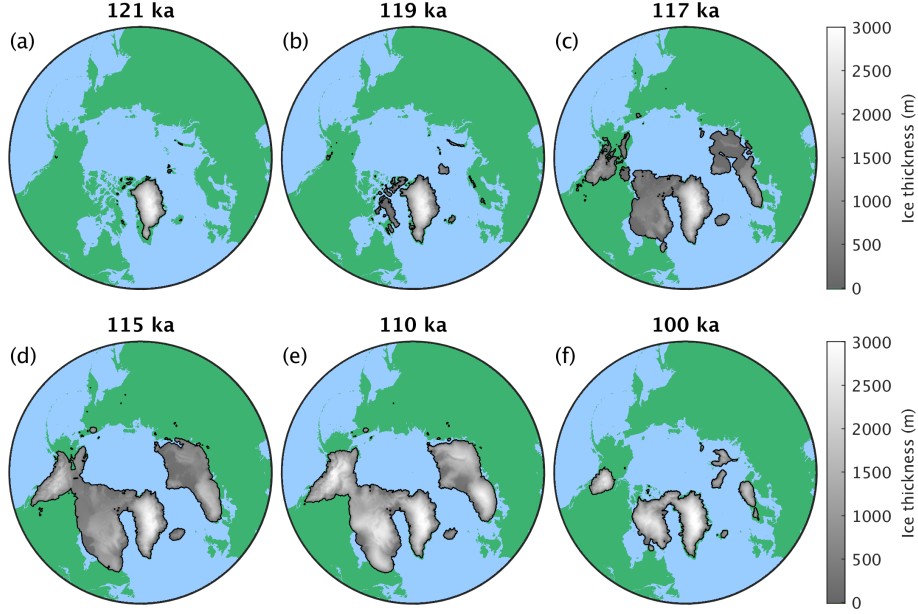

**Figure 6.** Ice sheet extent and thickness at different points in time for the reference last glacial inception simulation.

due to gravitational changes and due to the mass conservation between ice and ocean. Accordingly, the RSL rise in the loaded regions is dominated by the subsidence of the Earth surface due to the viscoelastic response of the solid Earth, whereas the

drop over the oceans reflects the mass conservation between ice and ocean, meaning the water equivalent of the ice sheet mass.



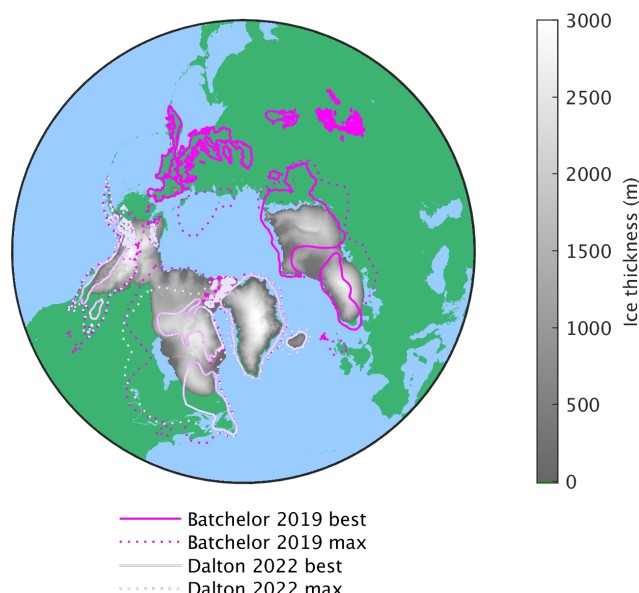

**Figure 7.** Simulated ice sheet extent at 110 ka (MIS5d) compared to the best and maximum extent reconstructions from Dalton et al. (2022) and Batchelor et al. (2019).

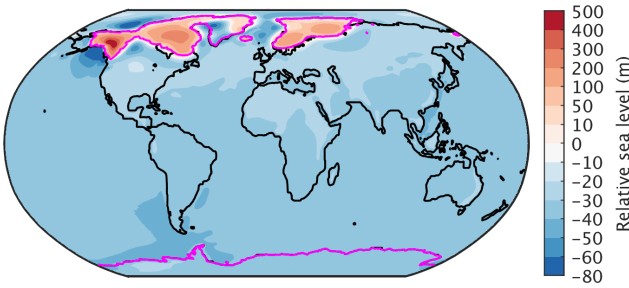

**Figure 8.** Simulated relative sea level changes relative to pre-industrial at 110 ka. The magenta lines indicate the ice sheet extent.

Furthermore, the ice sheets are surrounded by regions of lower RSL due the forebulge of the flexed elastic lithosphere. The bedrock deformation/depression is largest at the centers of the ice sheets, corresponding to a RSL that is up to 500 m higher than pre-industrial, while the far-field RSL is ∼35 m lower than pre-industrial (Fig. 8). Alaska shows the highest RSL values, which can be attributed to the low viscosity structure of the solid Earth in this region allowing a faster rebound in comparison
to the cratonic areas of NE Canada and Scandinavia (see viscosity structure in Fig. D1 in Appendix D).

In terms of changes in climate, from 125 ka to 110 ka the model shows a pronounced summer cooling at mid to high northern latitudes (Fig. 9c), as a direct response to the decrease in summer insolation (Fig. 9a), amplified by a strong albedo increase (Fig. 9b), which is driven by a combination of expanding sea ice, southward shift of the tree line (Fig. 10) and the establishment

 

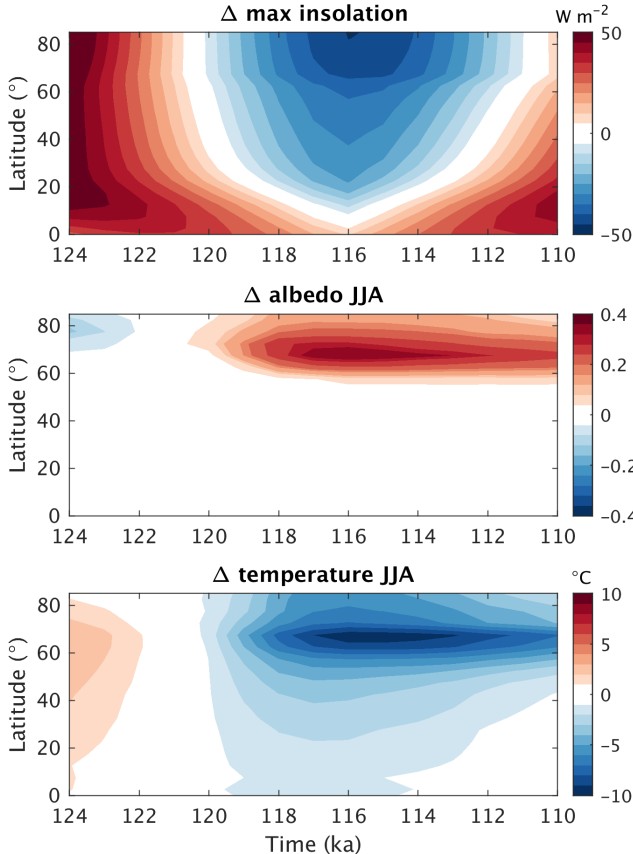

**Figure 9.** Zonal mean differences in (a) maximum summer insolation, (b) summer (JJA) surface albedo and (c) summer (JJA) surface air temperature as a function of time for the reference glacial inception simulation relative to the pre-industrial.

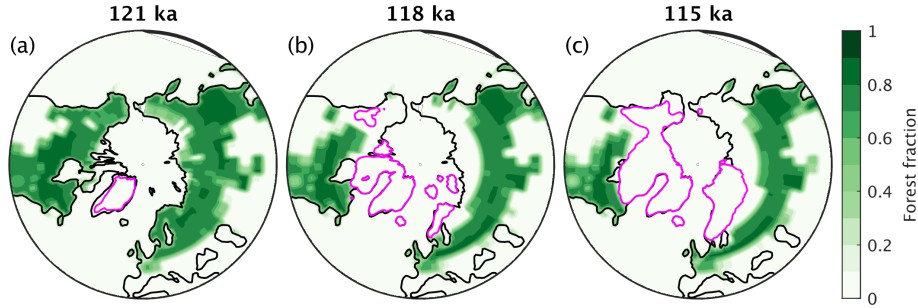

**Figure 10.** Forest fraction at different points in time simulated by the model in the reference experiment. The magenta lines indicate the ice sheet extent.

of land ice. Going from 121 ka to 115 ka, summer temperatures undergo a substantial cooling over most of the NH, particularly
over land (Fig. 11).




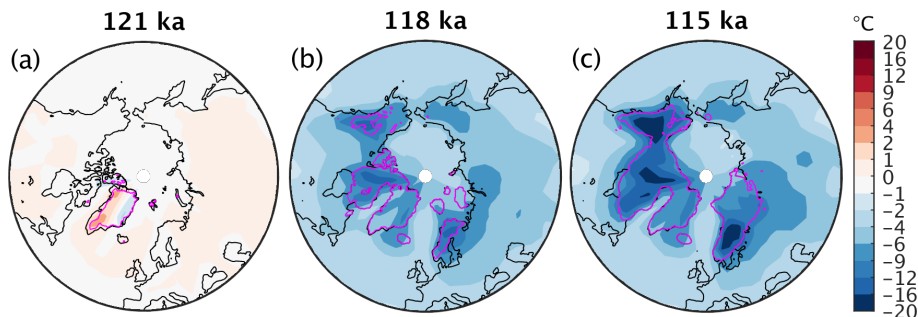

**Figure 11.** Summer (JJA) temperature difference relative to the pre-industrial at different times in the reference glacial inception simulations. Note the non-linear color scale.

## 4.2 Role of vegetation, ice-sheet and carbon-cycle feedbacks

The vegetation feedback plays a crucial role for glacial inception in our simulations. This is clearly illustrated by the much smaller increase in ice sheet area and volume in simulations where vegetation is prescribed at its equilibrium pre-industrial state, compared to the reference simulation with interactive vegetation (Fig. 12). Practically, fixed vegetation in our simulation means that the plant functional type fractions are not allowed to change and that the maximum leaf area index is fixed. However, for deciduos plants, the seasonality of the phenology will still be affected by the changing climatic conditions.

The effect of dynamic vegetation on climate can be isolated from two simulations, one with prescribed vegetation and one with dynamic vegetation, both without interactive ice sheets. The strong vegetation feedback is explained by a pronounced southward shift of the northern boreal treeline as a response to the gradual decrease in summer insolation at high northern latitudes during glacial inception (Fig. 13a-c). The expansion of tundra at the expense of taiga results in a substantial increase in surface albedo during the snow-covered season (Fig. 13d-f) with a particularly strong associated cooling in spring, which also extends into the summer months (Fig. 13g-i). This explains why the ice extent in the simulation with prescribed vegetation is generally reduced compared to the interactive vegetation case (Fig. 14a,b).

Our results on the important role played by the vegetation feedback for the initiation of NH glaciation are consistent with a number of previous studies that have shown that the vegetation response during the last glacial inception amplifies the orbitally induced summer cooling in high-northern latitudes (Harvey, 1989; De Noblet et al., 1996; Meissner et al., 2003; Crucifix and Loutre, 2002; Yoshimori et al., 2002), thus favoring the growth of ice sheets (Kageyama et al., 2004; Calov et al., 2005b; Mysak, 2008; Kubatzki et al., 2006).

The effect of ice sheets on glacial inception can be quantified with a simulation in which the land–ocean–ice sheet mask and the topography are prescribed at their pre-industrial state. This is equivalent to disabling the back-coupling of the ice sheets to the climate model and therefore suppresses the ice sheet feedback on climate. The model results indicate that this ice sheet feedback plays only a minor role during the ice growth phase until ∼115 ka (Fig. 12). This is explained by the ice expansion being driven by a rapid increase of perennially snow-covered area rather than by a slow lateral expansion of ice





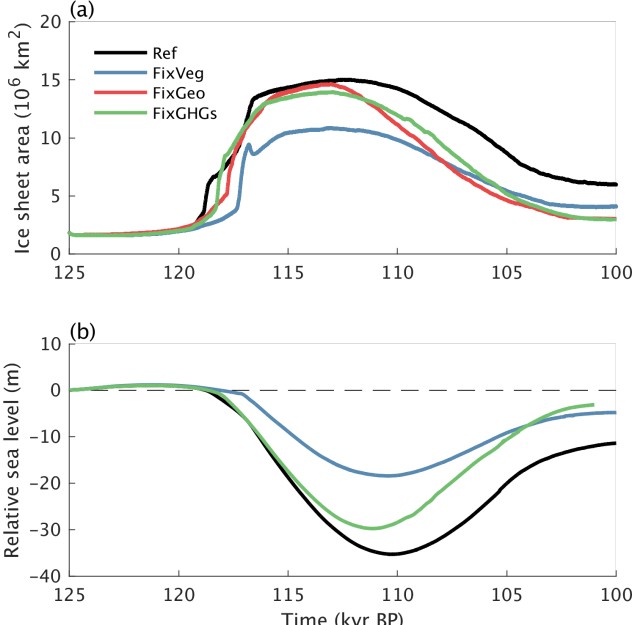

**Figure 12.** (a) Ice sheet area and (b) relative sea level changes for model simulations with fixed vegetation (blue), fixed present-day topography and land–ocean–ice sheet mask (red) and fixed 125 ka greenhouse gases concentrations (green) compared to the reference simulation (black).

sheets. However, the higher albedo of ice compared to ice-free land plays an important role in slowing down the ice sheet melt
during the ice retreat phase following the rising summer insolation after 115 ka (Fig. 12).

The role of the carbon cycle feedback during glacial inception can be estimated from a simulation where the atmospheric concentrations of the greenhouse gases are kept constant at their 125 ka values. Since the GHGs concentrations show only small variations until ∼115 ka (Fig. 4b), it is not surprising that the GHGs forcing, and therefore the carbon cycle feedback, plays only a minor role during the first phase of glacial inception (Fig. 12). Hence, the simulated ice sheets in the experiment with
prescribed constant GHGs are very similar to those in the reference simulation at 115 ka (Fig. 14a,c). However, the decrease in the equivalent $CO_2$ concentration after ∼115 ka, is important for slowing down the ice sheet melt and limit deglaciation (Fig. 12).

## 4.3 Sensitivity to climate model biases

It is known that ice sheets can be highly sensitive to relatively small temperature changes. For instance, it has been shown
that the bifurcation point for the complete melt of the Greenland ice sheet could be at only a few degrees above pre-industrial (Robinson et al., 2012; Höning et al., 2023). We therefore decided to apply different uniform temperature offsets in the surface energy and mass balance model and use these for sensitivity tests. This is justified because the global mean surface air temperature is (i) quite uncertain and (ii) different state-of-the-art climate models produce very different global mean temperatures



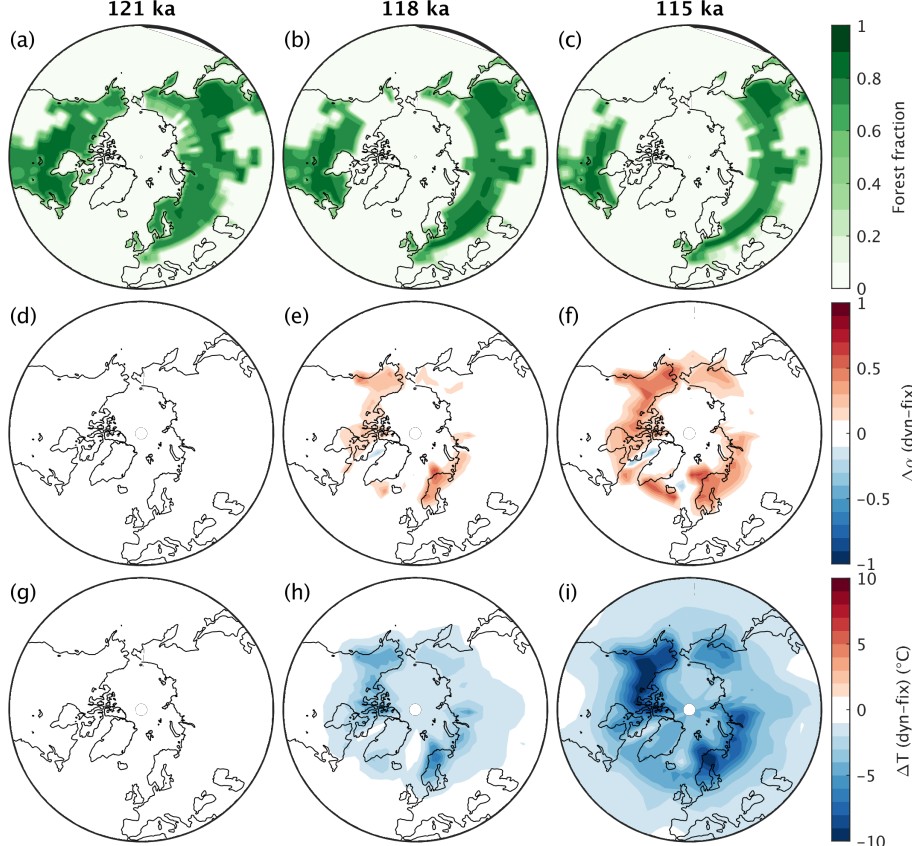

**Figure 13.** (a-c) Simulated forest fraction at different times during the glacial inception simulation with dynamic vegetation but prescribed present-day ice sheets (simulation NoIce). Differences in (d-f) surface albedo and (g-i) surface air temperature in late spring-early summer (May-June-July) between simulations with prescribed present-day ice sheets and dynamic (NoIce) and fixed (NoIceFixVeg) vegetation at different times.

(Bock et al., 2020). Our simulations show that the last glacial inception is sensitive to relatively small temperature perturbations in the surface mass balance model (Fig. 15). In particular, the difference in simulated sea level decrease between the experiment with a uniform cooling of $1\,°C$ and the experiment with a uniform warming of $1\,°C$ in SEMIX is $\sim$35 m (Fig. 15).

Without the dipole temperature correction over North America (Fig. B1), the simulated ice sheet distribution over this continent is shifted from the east to the west, as expected (Fig. 14a,d). Little ice is simulated in the area around the Hudson Bay, while ice extends further in the north-west. A similar east-west displacement of the ice distribution has also been found in other models that share temperature biases similar to the CLIMBER-X ones over North America (e.g. Bahadory et al., 2021).





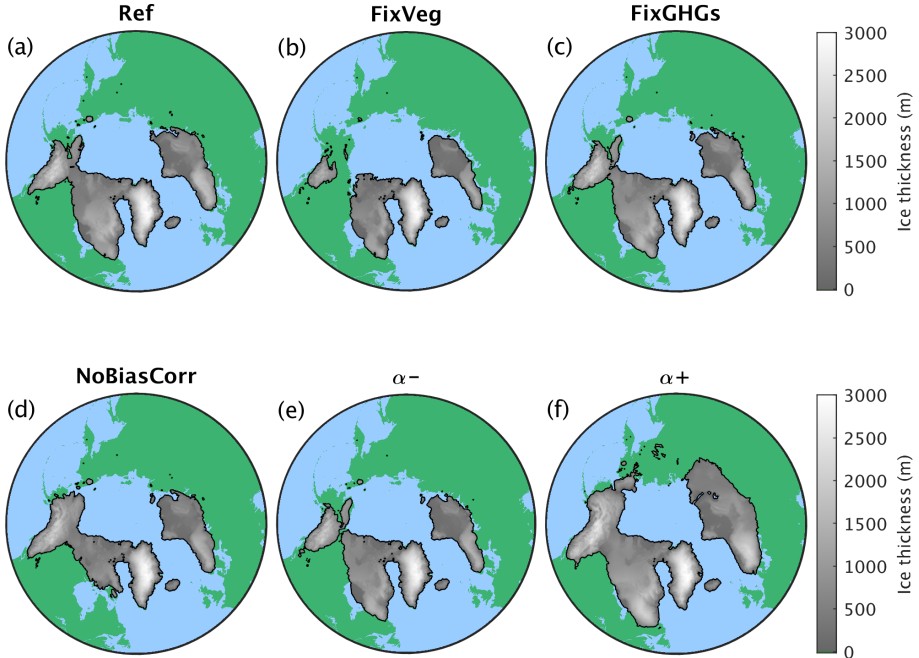

**Figure 14.** Simulated ice sheet extent and thickness at 115 ka for different experiments: (a) reference, (b) fixed pre-industrial vegetation, (c) fixed 125 ka GHGs concentrations, (d) no dipole temperature bias correction over North America, (e) reduced snow albedo and (f) increased snow albedo.

## 4.4 Sensitivity to snow albedo parameterisation

The albedo of snow is a function of snow grain size, with smaller grain sizes resulting in higher albedo (e.g. Warren and Wiscombe, 1980a; Gardner and Sharp, 2010). Fresh dry snow has a generally small grain size, but the grain size tends to increase as the snow undergoes metamorphism processes and, in particular, as melting occurs. Snow albedo is also affected by

the presence of light-absorbing impurities, such as minreal dust or soot particles (e.g. Dang et al., 2015; Warren and Wiscombe, 1980b). Since the albedo of snow is generally high, small relative changes in snow albedo will result in large relative changes in co-albedo, which is the relevant quantity determining the amount of absorbed solar radiation at the surface. It is therefore expected that uncertainties in the parameterisation of the albedo of snow will result in substantial differences in the surface energy and mass balance (e.g. Willeit and Ganopolski, 2018). To explore this we performed simulations in which we added

a constant offset of $\pm 0.025$ to the albedo of snow and experiments with half and double the dust deposition rate in SEMIX. These perturbations to the snow albedo result in a simulated sea level decrease during MIS5d that varies by more than $40\,\mathrm{m}$ (Fig. 16). This is remarkable, considering that the snow albedo changes introduced are of the order of only a few percent. In terms of spatial pattern, the simulated ice sheet is more extended and twice larger by volume if the snow albedo is higher and less extended if the snow albedo is lower than in the reference run (Fig. 14e,f).



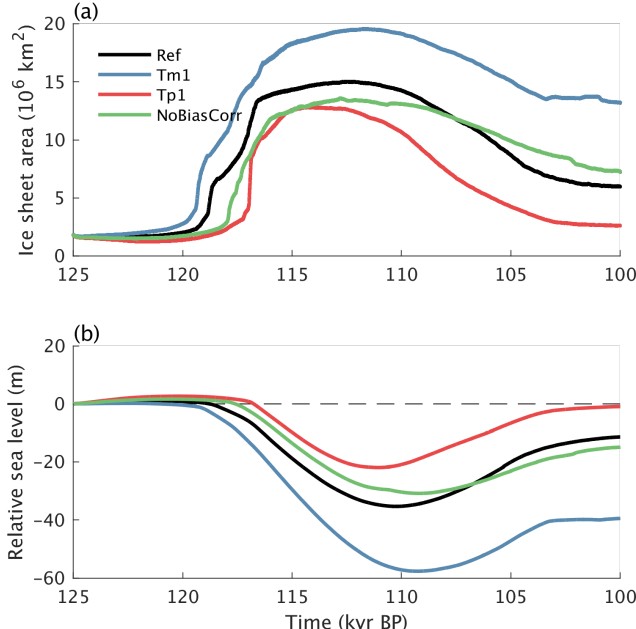

**Figure 15.** Simulated (a) ice sheet area and (b) relative sea level for different uniform temperature offsets in SEMIX (experiments Tp1 and Tm1 in Table 1) and for the simulation without temperature bias correction (NoBiasCorr).

## 4.5 Sensitivity to climate model acceleration

Several efforts are ongoing to simulate the last glacial cycle with state-of-the-art Earth system models based on general circulation models (e.g. Latif et al., 2016). Computational time is a strong constraint for these models and acceleration techniques (e.g. Lorenz and Lohmann, 2004) are one possible way to alleviate this problem. These are based on the fact that typical time scales of the atmosphere, ocean and vegetation are much shorter than orbital time scales. This allows to artificially accelerate the external forcings, in this case orbital parameters and GHGs concentration, which, considering that the atmosphere and ocean are also the most computationally expensive parts of an Earth system model, results in an effective speed-up of the model. The ice sheet model cannot be accelerated in the same way, because the time scale of ice sheets is comparable with the orbital time scales. But this is not problematic, because ice sheet models are comparatively less computationally demanding.

To assess the applicability of the acceleration technique, we perform additionally several experiments with different accelerations rates to explore how sensitive the ice sheet evolution during glacial inception is to the applied acceleration factor. Our results show only a relatively weak sensitivity of the ice sheet evolution to climate acceleration for acceleration factors up to ∼10 (Fig. 17), confirming earlier results from the CLIMBER-2 model (Calov et al., 2009; Ganopolski et al., 2010). Much larger accelerations factors don't allow for a proper representation of the positive climate feedbacks at work during glacial inception, resulting in reduced simulated ice extent and volume (Fig. 18).





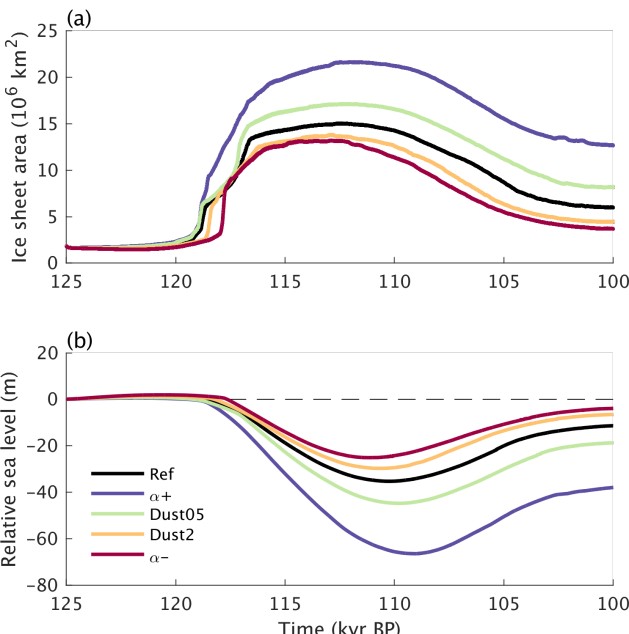

**Figure 16.** (a) Ice sheet area and (b) relative sea level changes for model simulations with scaled dust deposition fields and offsets in snow albedo.

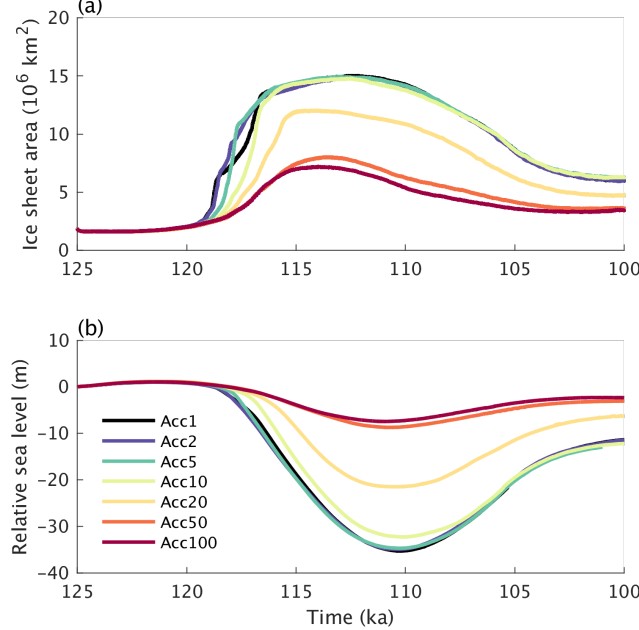

**Figure 17.** Simulated (a) ice sheet area and (b) relative sea level when different climate model acceleration factors are applied.





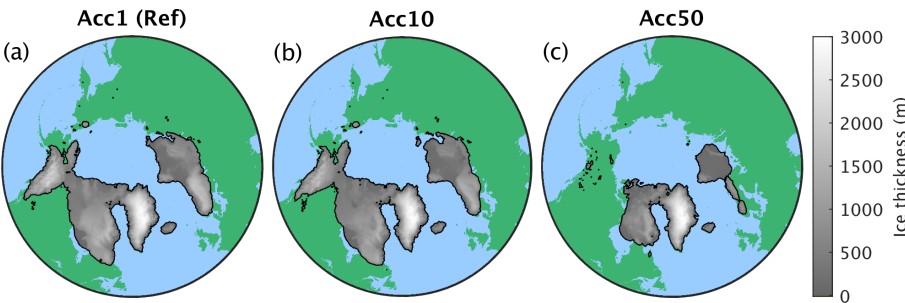

**Figure 18.** Simulated ice sheet extent and thickness at 115 ka for simulations with different climate acceleration factors: (a) reference run without climate acceleration, (b) acceleration factor 10 and (c) acceleration factor 50.

### 4.6 Sensitivity to ice sheet model resolution

We also tested the dependence of the simulated glacial inception on the resolution of the ice sheet model. A higher resolution of the ice sheet model will result in a better preservation of the fine-scale topographic structure and since the surface mass balance is strongly dependent on surface elevation, it is expected that better resolving mountain peaks would facilitate the formation of ice caps. It is unclear however, whether this would facilitate the formation of large scale ice sheets or not, also because better resolving mountains also implies that deep valleys are better resolved, which would inhibit the spreading of ice from isolated ice caps to larger scale ice sheets.

Our simulations show only a weak dependence of the model results on the resolution of the ice sheet model in the tested range between 16 km and 64 km (Fig. 19). However, some local ice caps that are formed with a high resolution ice sheet model are not resolved when the resolution is decreased (Fig. 20).

### 5 Discussion and conclusions

We have presented the results of a set of transient simulations of the last glacial inception with the CLIMBER-X Earth system model, which includes an ice sheet model and a model for the solid Earth response to changes in ice sheet loading. This paper also describes the ice sheet coupling with the atmosphere and ocean, which will serve as a reference for future studies using the model with interactive ice sheets.

We have shown that, as a response to the decreasing summer insolation at high northern latitudes, the model simulates a rapid expansion of ice sheets over northern North America and Scandinavia between ~120 ka and ~116 ka. This result is fully consistent with the concept of glacial inception as a bifurcation in the climate system (Calov et al., 2005a). The rapid expansion of the ice sheets' area was followed by ice volume growth reaching nearly 40 msle at 110 ka, which is in reasonable agreement with paleoclimate reconstructions. The albedo feedback associated with an increase in snow-covered area, sea ice extent and the southward retreat of the boreal forest plays a crucial role in the rapid ice area expansion. The vegetation feedback alone increases the maximum simulated ice sheet area by 50 %. On the other hand, the ice-sheet and carbon cycle feedbacks





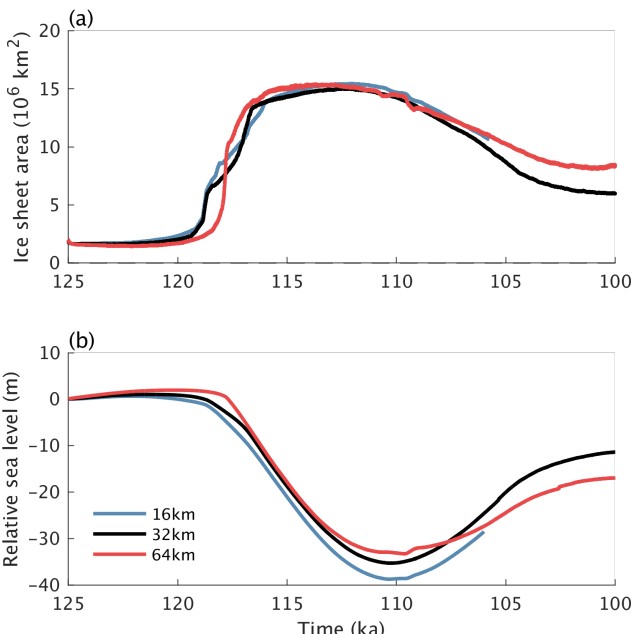

**Figure 19.** Simulated (a) ice sheet area and (b) relative sea level for different resolutions of the ice sheet model.

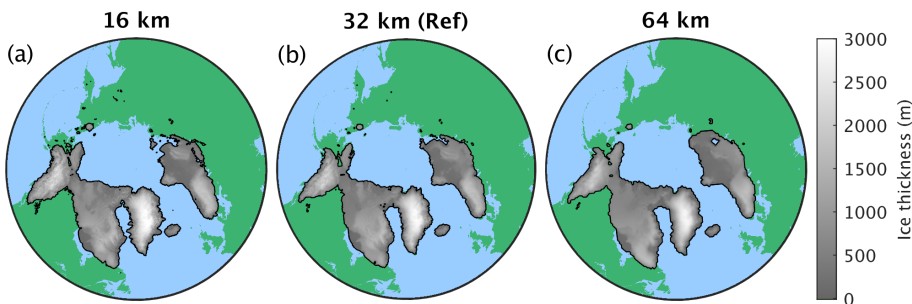

**Figure 20.** Simulated ice sheet extent and thickness at 115 ka for simulations with different horizontal resolutions of the ice sheet model: (a) 16 km, (b) 32 km (Ref) and (c) 64 km.

are of minor importance during the ice growth phase associated with decreasing summer insolation prior to ∼115 ka, but are fundamental in maintaining the system in a glacial state during the subsequent period of increase in summer insolation, resulting in an only partial deglaciation.

The results of model simulations demonstrate that the reduction of climate biases (too high summer air temperatures over eastern North America) leads to significant improvements in the simulated spatial extent of the North American ice sheet. Modeling results show strong sensitivity of simulated ice sheet evolution to the parameterization of clean snow albedo and to the impact of impurities on snow albedo.



The model results are not very sensitive to climate acceleration up to a factor ∼10. A climate acceleration factor of 10

would allow more complex Earth system models to run transient glacial inception simulations in a reasonable time using less

computational resources. The resolution of the ice sheet model does only marginally affect the model results, at least in the

tested range between 16 and 64 km, That is because a large-scale ice expansion over relatively flat terrain is the dominant

mechanism leading to glacial inception in our model, while ice caps, which can be captured only if the topography is highly

resolved, have only a very localized effect and are therefore not of fundamental importance.

The glacial inception simulations presented here are a first step towards simulating the full last glacial cycle with CLIMBER-

X.

*Code and data availability.* The source code of the climate component of CLIMBER-X v1.0 as used in the simulations of this paper is

archived on Zenodo (https://doi.org/10.5281/zenodo.7898797). The output of the model simulations is available on request from the con-

tributing authors.

**Appendix A: SICOPOLIS ice sheet model**

For the inclusion into CLIMBER-X, the original SICOPOLIS code has been restructured and organized into Fortran 90 derived

types to allow running several instances of the model at the same time, one for each ice sheet domain. Although in the present

study we employ a single ice sheet domain for the NH, CLIMBER-X can be set up to run with an arbitrary number of ice sheet

model domains, with potentially different resolutions, simply by specifying a list of domain names in a namelist.

As already mentioned above, SICOPOLIS is based on the shallow ice approximation for grounded ice, the shallow shelf

approximation for floating ice and hybrid shallow-ice–shelfy-stream dynamics for ice streams (Bernales et al., 2017), and the

enthalpy method of Greve and Blatter (2016) is used as thermodynamics solver.

In this study we use a Weertman-type sliding law to relate basal drag ($\tau_\mathrm{b}$) to basal velocity ($u_\mathrm{b}$):

$$\tau_\mathrm{b} = \frac{1}{c_\mathrm{s}^{1/p}} N^{q/p} u_\mathrm{b}^{1/p}, \tag{A1}$$

with $p = 3$ and $q = 2$. $N$ is the reduced basal pressure computed as the difference between the ice overburden pressure and the

water pressure if the base of the ice sheet is below sea level:

$$N = P_\mathrm{i} - P_\mathrm{w} = \rho_\mathrm{i} g h_\mathrm{i} - \rho_\mathrm{w} g \max(0, z_\mathrm{bed}), \tag{A2}$$

where $\rho_\mathrm{i}$ and $\rho_\mathrm{w}$ are the ice and water densities, $g$ is the acceleration due to gravity, $h_\mathrm{i}$ is ice thickness and $z_\mathrm{bed}$ is the bedrock

depth below sea level. The $c_\mathrm{s}$ sliding parameter depends on the assumed sediment fraction in each grid cell:

$$c_\mathrm{s} = (1 - f_\mathrm{sed}) c_\mathrm{s}^\mathrm{rock} + f_\mathrm{sed} c_\mathrm{s}^\mathrm{sed}. \tag{A3}$$

The sediment fraction is taken to be a linear function of sediment thickness between two critical values $h_\mathrm{sed}^\mathrm{min}$ and $h_\mathrm{sed}^\mathrm{max}$:

$$f_\mathrm{sed} = \frac{h_\mathrm{sed} - h_\mathrm{sed}^\mathrm{min}}{h_\mathrm{sed}^\mathrm{max} - h_\mathrm{sed}^\mathrm{min}}. \tag{A4}$$





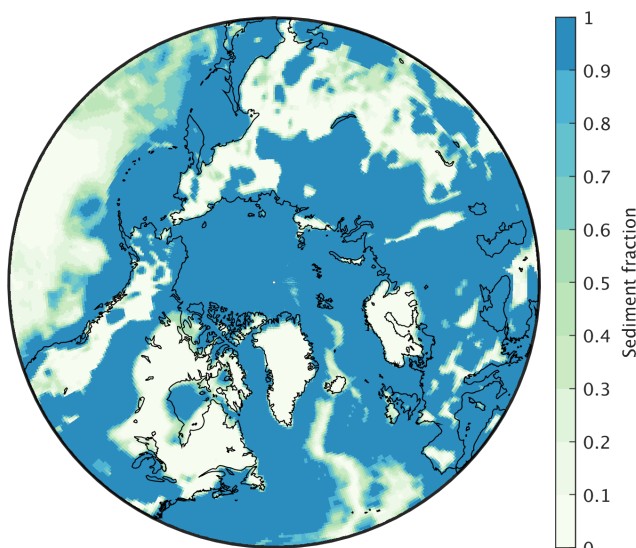

**Figure A1.** Sediment fraction used in the computation of the basal sliding coefficient $c_s$.

$f_{sed}$ is computed based on sediment thickness data from Laske et al. (2013) and is shown in Fig. A1. The value for bedrock is set to $c_s^{rock}=25\,\mathrm{myr}^{-1}\mathrm{Pa}^{-1}$ following the tuning for the Greenland ice sheet in Calov et al. (2018), while the value over thick sediments is simply taken as 10 times this value, i.e. $c_s^{sed}=250\,\mathrm{myr}^{-1}\mathrm{Pa}^{-1}$.

A broader investigation of the sliding law will be performed in forthcoming papers discussing the simulation of glacial termination, when the sliding is expected to be more important.

Iceberg calving at the marine-terminating ice front is parameterized using a simple thickness-based threshold method in which the calving rate is computed as:

$$C = \frac{h_{ice}^c - h_{ice}}{\tau_c}. \tag{A5}$$

It is applied only where the ice thickness, $h_{ice}$, is lower than the critical thickness for calving, $h_{ice}^c$, also in all neighboring ice points. The critical thickness for calving varies spatially and increases linearly with the depth of bedrock:

$$h_{ice}^c = h_{ice}^{c,deep} + \left( z_{bed}^{fil} - z_{bed}^{deep} \right) \cdot \frac{h_{ice}^{c,deep} - h_{ice}^{c,shallow}}{z_{bed}^{deep} - z_{bed}^{shallow}}, \tag{A6}$$

where $z_{bed}^{fil}$ is the bedrock elevation filtered with a Guassian filter with a $100\,\mathrm{km}$ radius and $z_{bed}^{shallow}$, $z_{bed}^{deep}$, $h_{ice}^{c,shallow}$ and $h_{ice}^{c,deep}$ are model parameters (Table A1). Although this is only a very crude representation, it is arguably more reasonable than assuming a uniform threshold which does not account for the different ocean dynamics in the narrow channels of the Canadian Arctic Archipelago and the deep open ocean. Lower values of $h_{ice}^c$ allow for a more rapid expansion of floating ice shelfs, thereby also affecting the rate of shelf ice spreading. However, since floating ice is usually thin, this has only a minor impact on the simulated ice volume during glacial inception.



**Table A1.** List of SICOPOLIS model parameters.

| Parameter | Value | Description |
|---|---|---|
| $c_s^{\text{rock}}$ | $25\,\text{m}\,\text{yr}^{-1}\text{Pa}^{-1}$ | sliding coefficient for rocks |
| $c_s^{\text{sed}}$ | $250\,\text{m}\,\text{yr}^{-1}\text{Pa}^{-1}$ | sliding coefficient for sediments |
| $h_{\text{sed}}^{\text{min}}$ | $10\,\text{m}$ | sediment thickness below which bare rock is assumed |
| $h_{\text{sed}}^{\text{max}}$ | $500\,\text{m}$ | sediment thickness above which full sediment cover is assumed |
| $z_{\text{bed}}^{\text{shallow}}$ | $200\,\text{m}$ | shallow water depth for calving |
| $z_{\text{bed}}^{\text{deep}}$ | $1000\,\text{m}$ | deep water depth for calving |
| $h_{\text{ice}}^{\text{c,shallow}}$ | $50\,\text{m}$ | critical calving thickness in shallow waters |
| $h_{\text{ice}}^{\text{c,deep}}$ | $500\,\text{m}$ | critical calving thickness in deep waters |
| $\tau_c$ | $10\,\text{yr}$ | time scale for calving |

The global geothermal heat flux product of Lucazeau (2019), substituted with data from Colgan et al. (2022) over Greenland, is applied as bottom boundary condition for the bedrock temperature equation at a depth of $2\,\text{km}$ below the land surface in SICOPOLIS.

   For all simulations presented in this study we used an annual time step for the thermodynamic part and a half-yearly time step for the dynamics.

## Appendix B:  SEMIX surface energy and mass balance interface


SEMIX is an adaptation of the surface energy and mass balance interface (SEMI) (Calov et al., 2005a) to CLIMBER-X. Its purpose is to determine the surface boundary conditions, namely surface mass balance and surface ice temperature, for the ice sheet model. In order to do that SEMIX has to bridge the gap in resolution between the climate model and the ice sheet model, which is achieved through a downscaling of the climate variables (Appendix B1). The surface energy and mass balance equations are then solved on the high-resolution ice sheet grid as outlined in Appendix B3. Since multiple ice sheet domains are allowed in CLIMBER-X, similarly to the ice sheet model also SEMIX can run in several separate instances, according to the defined ice sheet domains. SEMIX is called every $10\,\text{yr}$ over a full year with a time step of $1\,\text{day}$.


### B1 Downscaling of climate variables

SEMIX is driven by climate fields computed by the atmospheric model SESAM (Willeit et al., 2022). Since SESAM and SEMIX operate on very different grids, the first step required in the SEMIX coupling is the mapping of the climate fields from the coarse resolution, regular lat-lon SESAM grid onto the cartesian coordinates on the stereographic plane where the ice sheet model, and consequently SEMIX, operates. The mapping is done by simple bilinear interpolation using the four closest






**Table B1.** List climate model variables needed by SEMIX.

| Variable | Description | Unit |
|---|---|---|
| $z_\mathrm{s}$ | grid cell mean elevation | m |
| $\sigma_\mathrm{z}$ | standard deviation of sub-grid surface elevation over land | m |
| $T_\mathrm{atm}$ | atmospheric temperature | K |
| $r_\mathrm{atm}$ | atmospheric relative humidity | |
| $P$ | total precipitation rate | $\mathrm{kg\,m^{-2}\,s^{-1}}$ |
| $U$ | surface wind speed | $\mathrm{m\,s^{-1}}$ |
| $D$ | dust deposition rate | $\mathrm{kg\,m^{-2}\,s^{-1}}$ |
| $f_\mathrm{cld}$ | cloud cover fraction | |
| $\mathrm{SW}^\downarrow_\mathrm{TOA}$ | downward shortwave radiation at TOA | $\mathrm{W\,m^{-2}}$ |
| $\mu$ | daily mean cosine of solar zenith angle | rad |
| $\mathrm{SW}^\downarrow_\mathrm{VIS,dir}$ | downward shortwave visible radiation at surface, clear sky | $\mathrm{W\,m^{-2}}$ |
| $\mathrm{SW}^\downarrow_\mathrm{NIR,dir}$ | downward shortwave near-infrared radiation at surface, clear sky | $\mathrm{W\,m^{-2}}$ |
| $\mathrm{SW}^\downarrow_\mathrm{VIS,dif}$ | downward shortwave visible radiation at surface, cloudy sky | $\mathrm{W\,m^{-2}}$ |
| $\mathrm{SW}^\downarrow_\mathrm{NIR,dif}$ | downward shortwave near-infrared radiation at surface, cloudy sky | $\mathrm{W\,m^{-2}}$ |
| $\alpha_\mathrm{VIS,dir}$ | surface albedo for visible radiation at surface, clear sky | |
| $\alpha_\mathrm{NIR,dir}$ | surface albedo for near-infrared radiation at surface, clear sky | |
| $\alpha_\mathrm{VIS,dif}$ | surface albedo for visible radiation at surface, cloudy sky | |
| $\alpha_\mathrm{NIR,dif}$ | surface albedo for near-infrared radiation at surface, cloudy sky | |
| $\frac{\partial \mathrm{SW}^\downarrow}{\partial \alpha}\big|_\mathrm{VIS,dir}$ | partial derivative of $\mathrm{SW}^\downarrow_\mathrm{VIS,dir}$ wrt $\alpha_\mathrm{VIS,dir}$ | $\mathrm{W\,m^{-2}}$ |
| $\frac{\partial \mathrm{SW}^\downarrow}{\partial \alpha}\big|_\mathrm{NIR,dir}$ | partial derivative of $\mathrm{SW}^\downarrow_\mathrm{NIR,dir}$ wrt $\alpha_\mathrm{NIR,dir}$ | $\mathrm{W\,m^{-2}}$ |
| $\frac{\partial \mathrm{SW}^\downarrow}{\partial \alpha}\big|_\mathrm{VIS,dif}$ | partial derivative of $\mathrm{SW}^\downarrow_\mathrm{VIS,dif}$ wrt $\alpha_\mathrm{VIS,dif}$ | $\mathrm{W\,m^{-2}}$ |
| $\frac{\partial \mathrm{SW}^\downarrow}{\partial \alpha}\big|_\mathrm{NIR,dif}$ | partial derivative of $\mathrm{SW}^\downarrow_\mathrm{NIR,dif}$ wrt $\alpha_\mathrm{NIR,dif}$ | $\mathrm{W\,m^{-2}}$ |
| $\frac{\partial \mathrm{SW}^\downarrow_\mathrm{NIR,dir}}{\partial z}$ | partial derivative of $\mathrm{SW}^\downarrow_\mathrm{NIR,dir}$ wrt height | $\mathrm{W\,m^{-3}}$ |
| $\frac{\partial \mathrm{SW}^\downarrow_\mathrm{NIR,dif}}{\partial z}$ | partial derivative of $\mathrm{SW}^\downarrow_\mathrm{NIR,dif}$ wrt height | $\mathrm{W\,m^{-3}}$ |
| $\mathrm{LW}^\downarrow$ | downward longwave radiation at the surface | $\mathrm{W\,m^{-2}}$ |
| $\frac{\partial \mathrm{LW}^\downarrow}{\partial z}$ | partial derivative of $\mathrm{LW}^\downarrow$ wrt height | $\mathrm{W\,m^{-3}}$ |
| $T^\mathrm{bias}_\mathrm{2m,JJA}$ | near-surface summer air temperature bias at present-day | K |

neighboring grid points. A list of the climate model variables needed by SEMIX is given in Table B1. In the following we will denote the climate variables mapped onto the ice sheet grid with a superscript *i*.

A second step involves downscaling of the climate fields to account for e.g. the difference in surface elevation between the coarse climate model and the high-resolution ice sheet grid. A constant temperature lapse rate is used to adjust the atmospheric



temperature to the actual surface elevation:

$$T_{\mathrm{atm}} = T_{\mathrm{atm}}^i + \Gamma \left( z_{\mathrm{s}} - z_{\mathrm{s}}^i \right). \tag{B1}$$

Following Abe-Ouchi et al. (2007) and Kapsch et al. (2021) we use a value of $\Gamma = -5\,\mathrm{K\,km^{-1}}$ for the lapse rate. The near-405 surface air temperature is computed as the average of atmospheric temperature and skin temperature:

$$T_{\mathrm{2m}} = 0.5 \cdot \left( T_{\mathrm{atm}} - T_{\mathrm{skin}} \right). \tag{B2}$$

$T_{\mathrm{2m}}$ is then used to separate total precipitation into rain and snow, with the fraction of precipitation falling as snow computed as:

$$f_{\mathrm{snow}} = \begin{cases} 1, & T_{\mathrm{2m}} \leq (T_0 - 5) \\ 0.1 \cdot (T_0 + 5 - T_{\mathrm{2m}}), & (T_0 - 5) < T_{\mathrm{2m}} < (T_0 + 5) \\ 0, & T_{\mathrm{2m}} \geq (T_0 + 5) \end{cases} \tag{B3}$$

with $T_0 = 273.15\,\mathrm{K}$. Snowfall and rainfall rate are then derived from total precipitation as:

$$P_{\mathrm{snow}} = f_{\mathrm{snow}} \cdot P^i, \tag{B4}$$

$$P_{\mathrm{rain}} = (1 - f_{\mathrm{snow}}) \cdot P^i. \tag{B5}$$

$$\tag{B6}$$

Near-surface air specific humidity is computed from near-surface relative humidity and specific humidity at saturation as in 415 climate component of CLIMBER-X (Willeit et al., 2022):

$$q_{\mathrm{2m}} = r_{\mathrm{2m}} \cdot q_{\mathrm{sat}}(T_{\mathrm{2m}}, p_{\mathrm{s}}), \tag{B7}$$

with $r_{\mathrm{2m}} = (r_{\mathrm{atm}} + r_{\mathrm{skin}})/2$, $r_{\mathrm{skin}} = q_{\mathrm{atm}}/q_{\mathrm{sat}}(T_{\mathrm{skin}}, p_{\mathrm{s}})$ and $q_{\mathrm{atm}} = r_{\mathrm{atm}}^i \cdot q_{\mathrm{sat}}(T_{\mathrm{atm}}, p_{\mathrm{s}})$.

The downward shortwave radiation fluxes at the surface on the ice sheet model grid are computed from the interpolated fluxes and adjusted using the partial derivatives of the radiation fields with respect to surface albedo and surface elevation:

$$\mathrm{420} \quad \mathrm{SW}_{\mathrm{VIS,dir}}^{\downarrow} = \mathrm{SW}_{\mathrm{VIS,dir}}^{\downarrow,i} + \left. \frac{\partial \mathrm{SW}^{\downarrow}}{\partial \alpha} \right|_{\mathrm{VIS,dir}}^i \left( \alpha_{\mathrm{VIS,dir}} - \alpha_{\mathrm{VIS,dir}}^i \right) \tag{B8}$$

$$\mathrm{SW}_{\mathrm{NIR,dir}}^{\downarrow} = \mathrm{SW}_{\mathrm{NIR,dir}}^{\downarrow,i} + \left. \frac{\partial \mathrm{SW}^{\downarrow}}{\partial \alpha} \right|_{\mathrm{NIR,dir}}^i \left( \alpha_{\mathrm{NIR,dir}} - \alpha_{\mathrm{NIR,dir}}^i \right) + \frac{\partial \mathrm{SW}_{\mathrm{NIR,dir}}^{\downarrow}}{\partial z}^i \left( z_{\mathrm{s}} - z_{\mathrm{s}}^i \right) \tag{B9}$$

$$\mathrm{SW}_{\mathrm{VIS,dif}}^{\downarrow} = \mathrm{SW}_{\mathrm{VIS,dif}}^{\downarrow,i} + \left. \frac{\partial \mathrm{SW}^{\downarrow}}{\partial \alpha} \right|_{\mathrm{VIS,dif}}^i \left( \alpha_{\mathrm{VIS,dif}} - \alpha_{\mathrm{VIS,dif}}^i \right) \tag{B10}$$

$$\mathrm{SW}_{\mathrm{NIR,dif}}^{\downarrow} = \mathrm{SW}_{\mathrm{NIR,dif}}^{\downarrow,i} + \left. \frac{\partial \mathrm{SW}^{\downarrow}}{\partial \alpha} \right|_{\mathrm{NIR,dif}}^i \left( \alpha_{\mathrm{NIR,dif}} - \alpha_{\mathrm{NIR,dif}}^i \right) + \frac{\partial \mathrm{SW}_{\mathrm{NIR,dif}}^{\downarrow}}{\partial z}^i \left( z_{\mathrm{s}} - z_{\mathrm{s}}^i \right). \tag{B11}$$





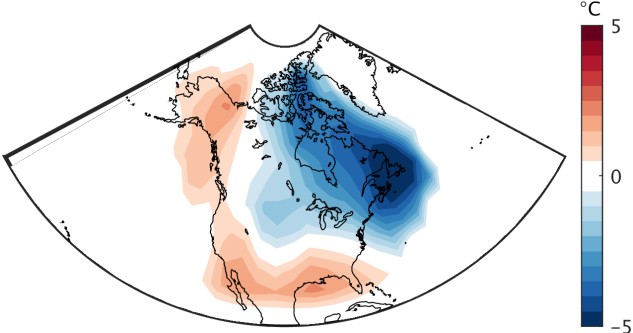

**Figure B1.** Temperature correction dipole applied in SEMIX.

The first correction term is needed because the downward shortwave radiation flux at the surface depends itself on the surface
albedo through its influence on multiple scattering between surface and atmosphere. A larger surface albedo will generally
result in a larger downward shortwave radiation flux. The elevation correction term for the near-infrared component arises from
the absorbtion of shortwave radiation by the atmosphere in the near-infrared band, which introduces an elevation dependence
of the radiative flux. Finally, the net shortwave flux absorbed at the surface, which is the quantity entering the surface energy
balance equation, is derived as a weighted average of clear sky (direct radiation) and cloudy sky (diffuse radiation) radiative
fluxes using cloud cover fraction:

$$
\begin{aligned}
\mathrm{SW}_{\mathrm{net}} = {} & \left(1 - f_{\mathrm{cld}}^i\right) \cdot \left(\mathrm{SW}_{\mathrm{VIS,dir}}^\downarrow \cdot (1 - \alpha_{\mathrm{VIS,dir}}) + \mathrm{SW}_{\mathrm{NIR,dir}}^\downarrow \cdot (1 - \alpha_{\mathrm{NIR,dir}})\right) \\
& + f_{\mathrm{cld}}^i \cdot \left(\mathrm{SW}_{\mathrm{VIS,dif}}^\downarrow \cdot (1 - \alpha_{\mathrm{VIS,dif}}) + \mathrm{SW}_{\mathrm{NIR,dif}}^\downarrow \cdot (1 - \alpha_{\mathrm{NIR,dif}})\right)
\end{aligned}
\tag{B12}
$$

A similar approach is applied also for the downscaling of the downward longwave radiation at the surface using the partial
derivative of the flux with respect to elevation:

$$
\mathrm{LW}^\downarrow = \mathrm{LW}^{\downarrow,i} + \frac{\partial \mathrm{LW}^{\downarrow\, i}}{\partial z} \left(z_{\mathrm{s}} - z_{\mathrm{s}}^i\right).
\tag{B13}
$$

### B2 Temperature bias correction

As described in Section 2, a temperature bias correction is applied only over North America. The bias correction is applied to
the atmospheric temperature and eq. B1 therefore becomes:

$$
T_{\mathrm{atm}} = T_{\mathrm{atm}}^i + \Gamma \left(z_{\mathrm{s}} - z_{\mathrm{s}}^i\right) - T_{\mathrm{2m,JJA}}^{\mathrm{bias}^i},
\tag{B14}
$$

where $T_{\mathrm{2m,JJA}}^{\mathrm{bias}^i}$ is the mean summer (June-July-August) temperature bias in CLIMBER-X relative to the ERA5 reanalysis
(Hersbach et al., 2020) over the time period from 1981 to 2010 and is shown in Fig. B1. The same temperature bias field is
applied at all time steps throughout the year.

Furthermore, since the downwelling longwave radiation at the surface, which also affects the surface energy balance, is
closely related to near-surface air temperature, we also correct the downwelling longwave radiation for the temperature bias



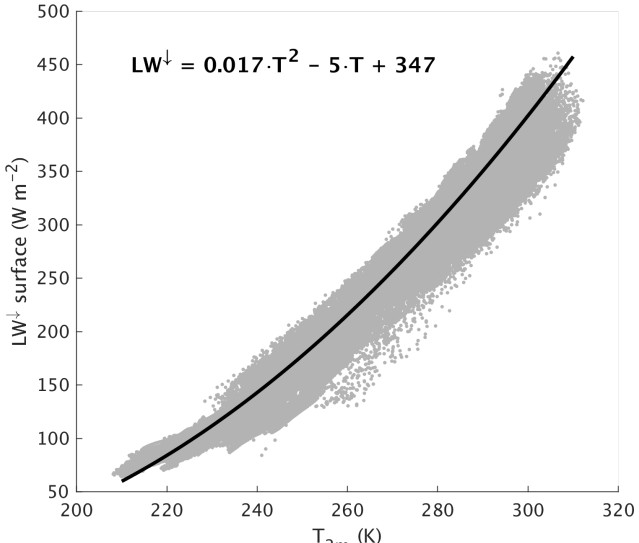

**Figure B2.**

using a simple quadratic relation derived from ERA5 reanalysis data (Fig.B2). Equation. B13 is then modified to:

$$\mathrm{LW}^{\downarrow} = \mathrm{LW}^{\downarrow,i} + \frac{\partial \mathrm{LW}^{\downarrow\,i}}{\partial z}\left(z_{\mathrm{s}} - z_{\mathrm{s}}^{i}\right) - (0.034 \cdot T_{\mathrm{2m}} - 5) \cdot T_{\mathrm{2m,JJA}}^{\mathrm{bias}^{i}}. \tag{B15}$$

### B3  Surface energy and mass balance of snow and ice

The surface energy balance in SEMIX is written as:

$$\mathrm{SW}_{\mathrm{net}} + \mathrm{LW}^{\downarrow} - \mathrm{LW}^{\uparrow} - \mathrm{SH} - \mathrm{LE} - G = 0, \tag{B16}$$

where $\mathrm{SW}_{\mathrm{net}}$ is the net shortwave radiation absorbed at the surface, $\mathrm{LW}^{\downarrow}$ and $\mathrm{LW}^{\uparrow}$ are the downwelling and upwelling longwave radiation at the surface, SH is the sensible heat flux, LE is the latent heat flux and $G$ the heat flux into the snow/ice. Equation (B16) is then solved for the skin temperature, $T_{\mathrm{skin}}$, using the formulations for the energy fluxes described next.

The surface albedo values used to compute the net shortwave at the surface in eq.(**??**) are defined as:

$$\alpha_{\mathrm{VIS,dir}} = f_{\mathrm{snow}} \cdot \alpha_{\mathrm{VIS,dir}}^{\mathrm{snow}} + (1 - f_{\mathrm{snow}}) \cdot \alpha^{\mathrm{bg}} \tag{B17}$$

$$\alpha_{\mathrm{NIR,dir}} = f_{\mathrm{snow}} \cdot \alpha_{\mathrm{NIR,dir}}^{\mathrm{snow}} + (1 - f_{\mathrm{snow}}) \cdot \alpha^{\mathrm{bg}} \tag{B18}$$

$$\alpha_{\mathrm{VIS,dif}} = f_{\mathrm{snow}} \cdot \alpha_{\mathrm{VIS,dif}}^{\mathrm{snow}} + (1 - f_{\mathrm{snow}}) \cdot \alpha^{\mathrm{bg}} \tag{B19}$$

$$\alpha_{\mathrm{NIR,dif}} = f_{\mathrm{snow}} \cdot \alpha_{\mathrm{NIR,dif}}^{\mathrm{snow}} + (1 - f_{\mathrm{snow}}) \cdot \alpha^{\mathrm{bg}}. \tag{B20}$$

$$\tag{B21}$$





where the snow cover fraction $f_{\text{snow}}$ is determined considering also the effect of topographic roughness following Niu and Yang (2007) and Roesch et al. (2001):

$$f_{\text{snow}} = \tanh\left(\frac{h_{\text{snow}}}{10 \cdot z_0}\right) \frac{h_{\text{snow}}}{h_{\text{snow}} + 2 \times 10^{-4} \cdot \sigma_z^i}. \tag{B22}$$

$h_{\text{snow}}$ is snow thickness and $z_0$ is the surface roughness length. The snow albedo scheme is the same as used in the climate component of CLIMBER-X and includes a dependence on snow grain size and dust and soot concentration following Dang et al. (2015). The background albedo is a weighted average between a constant bare soil albedo and variable ice albedo:

$$\alpha_{\text{bg}} = f_{\text{ice}} \cdot \alpha^{\text{ice}} + (1 - f_{\text{ice}}) \cdot \alpha^{\text{soil}}. \tag{B23}$$

The ice cover fraction is computed from ice sheet thickness, $h_{\text{ice}}$, and topographic roughness as:

$$f_{\text{ice}} = \tanh\left(\frac{h_{\text{ice}}}{\sigma_z^i}\right). \tag{B24}$$

For ice-free grid cells next to the ice sheet margin, where $h_{\text{ice}} = 0$, an ice thickness is computed instead as an average over the 3x3 grid cells neighorhood, and will therefore by definition be $> 0$. The ice albedo of newly formed ice is set to a constant value representative for firn, $\alpha^{\text{firn}} = 0.7$. As soon as ice starts to melt, the albedo gradually decreases towards a clean ice albedo, assuming that the clean ice albedo is reached when the firn layer is melted:

$$\frac{\Delta\alpha^{\text{ice}}}{\Delta t} = -\frac{\alpha^{\text{ice}} - \alpha^{\text{ice}}_{\text{clean}}}{h_{\text{firn}}\rho_{\text{firn}}/M_{\text{ice}}}, \tag{B25}$$

where $M_{\text{ice}}$ is the rate of ice melt, $h_{\text{firn}}$ is the constant thickness of the firn layer and $\rho_{\text{firn}}$ is the density of firn, which is also considered to be constant (Tab. B2). As the ice becomes snow-free and therefore exposed to the deposition of dust and other wind-born material that darken the ice, the albedo is assumed to decrease further towards a dirty ice albedo, $\alpha^{\text{ice}}_{\text{dirty}} = 0.4$, with a time scale of 100 yr. Whenever the annual surface mass balance is positive, indicating accumulation of snow and consequently formation of a firn layer, the ice albedo is reset to the albedo of firn ($\alpha^{\text{ice}} = \alpha^{\text{firn}}$). A few considerations are appropriate here on the representation of ice albedo in ice-free grid cells at the ice sheet margin. If the surface mass balance is positive in these points, the ice (or background) albedo is not very important, because it implies that not all snow is melted during summer and therefore the background does not become exposed. On the other hand, if the surface mass balance is negative, the background albedo in the ice-free margin points should reflect the properties of the ice that could flow into these points from neighboring grid cells through horizontal ice flow. In this case the ice albedo will determine how negative the mass balance is, through its control on the absorbed radiation when all snow is melted. How negative the surface mass balance is will eventually determine whether the ice flowing into these grid cells will be melted completely or not, and therefore plays a role for the velocity at which ice sheet cover can expand laterally. Based on these considerations the ice albedo in the grid cells at the ice sheet margin is computed as the average of the ice albedo of the neighoring ice points.

The surface emitted longwave radiation is given by the Stefan-Boltzmann law:

$$\text{LW}^\uparrow = \epsilon\sigma T^4_{\text{skin}}, \tag{B26}$$



with $\epsilon$ the surface emissivity and $\sigma$ the Stefan-Boltzmann constant.

The sensible heat flux is computed from the temperature gradient between the skin and near–surface air, using the bulk aerodynamic formula:

$$\text{SH} = \frac{\rho_{\text{a}} c_{\text{p}}}{r_{\text{aer}}} (T_{\text{skin}} - T_{\text{2m}}),\tag{B27}$$

where $\rho_{\text{a}}$ is air density, $c_{\text{p}}$ is the specific heat of air and $r_{\text{aer}}$ is the aerodynamic resistance. Similarly, the latent heat flux over sea ice is expressed in terms of the specific humidity gradient between the surface and near–surface air:

$$\text{LE} = L \frac{\rho_{\text{a}}}{r_{\text{aer}}} \left( q_{\text{sat}}(T_{\text{skin}}) - q_{\text{2m}} \right).\tag{B28}$$

$L$ is the latent heat of sublimation and $q_{\text{sat}}$ is the specific humidity at saturation. The aerodynamic resistance is computed from wind speed, surface exchange coefficient and bulk Richardson number following Willeit and Ganopolski (2016).

The conductive heat flux into the snow/ice ($G$) is computed as:

$$G = \frac{\lambda}{\Delta z_1} \left( T_{\text{skin}} - T_{\text{s},1} \right),\tag{B29}$$

where $\lambda$ is the heat conductivity of snow/ice and $\Delta z_1$ is the distance between the surface and the middle of the snow layer or uppermost ice layer with temperature $T_{\text{s},1}$, depending whether a snow layer is present or not.

The prognostic terms in $T_{\text{skin}}$ in the formulation of the surface energy fluxes are then linearized using Taylor series expansion assuming that the temperature at the new time step, $T_{\text{skin},n+1} = T_{\text{skin},n} + \Delta T_{\text{skin}}$ with $\Delta T_{\text{skin}} \ll T_{\text{skin}}$:

$$T_{\text{skin},n+1}^4 = T_{\text{skin},n}^4 + 4T_{\text{skin},n}^3(T_{\text{skin},n+1} - T_{\text{skin},n}),\tag{B30}$$


$$q_{\text{sat}}(T_{\text{skin},n+1}) = q_{\text{sat}}(T_{\text{skin}}) + \left. \frac{dq_{\text{sat}}}{dT_{\text{skin}}} \right|_{T_{\text{skin}}=T_{\text{skin},n}} (T_{\text{skin},n+1} - T_{\text{skin},n}).\tag{B31}$$

Equation (B16) can then be solved explicitly for the skin temperature at the new time step, $T_{\text{skin},n+1}$. If the skin temperature is above freezing the surface energy fluxes are diagnosed first with the skin temperature greater then $0\,°\text{C}$ and then with skin temperature set to $0\,°\text{C}$. The difference between the sum of the energy fluxes is then used to directly melt snow and/or ice, 510   without the need for the snow or ice temperature to be at melting point.

The heat transfer in the snow–ice column is represented by a one-dimensional heat diffusion equation. A single snow layer is represented on top of five unevenly spaced vertical layers in the ice reaching a depth of $15\,\text{m}$. The heat flux $G$ is applied as top boundary condition, while a no-flux condition is applied at the bottom of the ice column. If the temperature of the snow or ice layers is greater than $0\,°\text{C}$, the excess energy is used to melt snow or ice. Liquid water produced by snowmelt or added to 515   the snow layer from rainfall can be refrozen in the snow layer or refreeze to form superimposed ice. The liquid water available for refreezing is:

$$F_{\text{avail}} = (M_{\text{snow}} + P_{\text{rain}})\Delta t,\tag{B32}$$



where $M_{\mathrm{snow}}$ is snowmelt and $\Delta t$ the time step in SEMIX ($1\,\mathrm{day}$). The maximum amount of water that is allowed to refreeze is taken to be a linear function of the thickness of the snow layer and accounts for the amount of water that has already been
refrozen since the start of the melt season:

$$F_{\mathrm{max}} = \phi_{\mathrm{snow}} h_{\mathrm{snow}} \rho_{\mathrm{snow}} - \sum F, \tag{B33}$$

where $\phi_{\mathrm{snow}}$ is snow porosity, $\rho_{\mathrm{snow}}$ is snow density and $F$ is refreezing. The maximum amount of water available for refreezing then becomes:

$$F_{\mathrm{avail}}^{\mathrm{max}} = \min\left(F_{\mathrm{avail}}, F_{\mathrm{max}}\right). \tag{B34}$$

The amount of water that can potentially be refrozen depends on the 'cold content' of the snow layer:

$$F_{\mathrm{pot}} = \frac{C_{\mathrm{i}}}{L_{\mathrm{f}}} h_{\mathrm{snow}} \rho_{\mathrm{snow}} \cdot (T_0 - T_{\mathrm{snow}}), \tag{B35}$$

with $C_{\mathrm{i}}$ the specific heat capacity of ice, $L_{\mathrm{f}}$ the latent heat of fusion and $T_{\mathrm{snow}}$ the temperature of the snow layer. The actual refreezing rate is then simply the minimum between available and potential:

$$F = \frac{\min\left(F_{\mathrm{avail}}^{\mathrm{max}}, F_{\mathrm{pot}}\right)}{\Delta t}. \tag{B36}$$

A fraction $f_{\mathrm{rfz,snow}}$ is then assumed to refreeze within the snow layer, while the rest forms superimposed ice:

$$f_{\mathrm{rfz,snow}} = f_{\mathrm{rfz,snow}}^{\mathrm{max}} \cdot \min\left(1, \frac{h_{\mathrm{snow}}}{h_{\mathrm{snow}}^{\mathrm{crit}}}\right), \tag{B37}$$

where $f_{\mathrm{rfz,snow}}^{\mathrm{max}}$ and $h_{\mathrm{snow}}^{\mathrm{crit}}$ are model parameters (Table **??**).

Finally, the surface mass balance and runoff are diagnosed as:

$$\mathrm{SMB} = P_{\mathrm{snow}} - E - M_{\mathrm{snow}} - M_{\mathrm{ice}} + F \tag{B38}$$

$$R = M_{\mathrm{snow}} + M_{\mathrm{ice}} + P_{\mathrm{rain}} - F. \tag{B39}$$

These are then integrated over the whole year and passed to the ice sheet model. The surface ice temperature, which is needed by the ice sheet model as top boundary condition in the temperature equation, is computed as the annual average temperature of the deepest ice layer ($\sim 10\,\mathrm{m}$).

## Appendix C: IMO ice shelf basal melt model

In CLIMBER-X we have implemented the simple and general ice shelf basal melt parameterisations of Beckmann and Goosse (2003) and Pollard and Deconto (2012), both of which rely on the difference between the ambient water temperature derived from ocean or lake model and the freezing point temperature at the ice shelf base.

The basal mass balance in the linear model of Beckmann and Goosse (2003) is:

$$\mathrm{BMB} = k_1 \frac{\rho_{\mathrm{w}} C_{\mathrm{w}}}{\rho_{\mathrm{i}} L_{\mathrm{f}}} (T_{\mathrm{w}} - T_{\mathrm{f}}), \tag{C1}$$





**Table B2.** List of SEMIX model parameters.

| Parameter | Value | Description |
|---|---|---|
| $\Gamma$ | $5\,\mathrm{K\,km^{-1}}$ | lapse rate of surface temperature |
| $\alpha^{\mathrm{soil}}$ | 0.2 | bare soil albedo |
| $\alpha^{\mathrm{ice}}_{\mathrm{firn}}$ | 0.7 | firn albedo |
| $\alpha^{\mathrm{ice}}_{\mathrm{clean}}$ | 0.55 | clean ice albedo |
| $\alpha^{\mathrm{ice}}_{\mathrm{dirty}}$ | 0.4 | dirty ice albedo |
| $h_{\mathrm{firn}}$ | $100\,\mathrm{m}$ | thickness of firn layer |
| $\rho_{\mathrm{firn}}$ | $500\,\mathrm{kg\,m^{-3}}$ | firn density |
| $\rho_{\mathrm{snow}}$ | $250\,\mathrm{kg\,m^{-3}}$ | snow density |
| $\phi_{\mathrm{snow}}$ | 0.7 | snow porosity |
| $C_{\mathrm{i}}$ | $2110\,\mathrm{J\,kg^{-1}K^{-1}}$ | specific heat capacity of ice |
| $L_{\mathrm{f}}$ | $3.34\times10^5\,\mathrm{J\,kg^{-1}}$ | latent heat of fusion of ice |
| $f^{\mathrm{max}}_{\mathrm{rfz,snow}}$ | 0.5 | maximum fraction of refreezing in snow layer |
| $h^{\mathrm{crit}}_{\mathrm{snow}}$ | $4\,\mathrm{m}$ | critical snow thickness for refreezing |

where $k_1$ is a model parameter (Tab. C1), $\rho_{\mathrm{w}}$ is a typical seawater density, $C_{\mathrm{w}}$ is the specific heat capacity of water, $\rho_{\mathrm{i}}$ is the density of ice and $L_{\mathrm{f}}$ is the latent heat of fusion of ice. $T_{\mathrm{w}}$ is either the seawater temperature from the ocean model for marine-terminating margins or the lake-water temperature for lake-terminating margins, horizontally extrapolated to the ice sheet model grid and vertically interpolated to the depth of the ice shelf base. $T_{\mathrm{f}}$ is the freezing point temperature at the base of the ice shelf (Beckmann and Goosse, 2003):

$$T_{\mathrm{f}} = 0.0939 - 0.057 \cdot S_{\mathrm{w}} - 7.64 \times 10^{-4} \cdot z_{\mathrm{b}} \tag{C2}$$

where $S_{\mathrm{w}}$ is the water salinity, derived similarly to $T_{\mathrm{w}}$, assuming that the salinity of lakes is zero, and $z_{\mathrm{b}}$ is the depth of the ice shelf base below sea level.

The basal mass balance in the model of Pollard and Deconto (2012) is similar, but relies on a quadratic dependence on the temperature gradient:

$$\mathrm{BMB} = k_2 \frac{\rho_{\mathrm{w}} C_{\mathrm{w}}}{\rho_{\mathrm{i}} L_{\mathrm{f}}} \left| T_{\mathrm{w}} - T_{\mathrm{f}} \right| \cdot \left( T_{\mathrm{w}} - T_{\mathrm{f}} \right), \tag{C3}$$

where $k_2$ is a model parameter and all other terms in this equation have already been defined above.

Similarly to Quiquet et al. (2021), in order to avoid unrealistic ice shelf expansion over the deep ocean we apply an additional scaling of basal melt with the depth of the bedrock elevation:

$$f_{\mathrm{BMB}} = 1 + \frac{\max\left(0, z_{\mathrm{bed}} - z_{\mathrm{bed}}^{\mathrm{ref}}\right)}{z_{\mathrm{bed}}^{\mathrm{ref}}}, \tag{C4}$$





**Table C1.** List of IMO model parameters.

| Parameter | Value | Description |
|---|---|---|
| $k_1$ | $1 \times 10^{-4}$ | parameter for the Beckmann and Goosse (2003) model |
| $k_2$ | $5 \times 10^{-4}$ | parameter for the Pollard and Deconto (2012) model |
| $z_{\mathrm{bed}}^{\mathrm{ref}}$ | $200\,\mathrm{m}$ | critical bedrock depth for basal melt scaling |

with $z_{\mathrm{bed}}^{\mathrm{ref}}$ a model parameter (Tab. C1).

IMO is called every month to resolve the seasonal cycle in ocean/lake temperature, but the coupling with the ice sheet model is done yearly.

## Appendix D: VILMA solid Earth model

The solid Earth model VILMA solves the field equations of a viscoelastic incompressible and self-gravitating continuum in
the spherical domain for the mantle and lithosphere of the Earth. The fluid core as well as loading processes at the surface are considered as boundary conditions at the respective boundaries. Lateral changes in viscosity are considered and are following the formulation as an initial value problem discussed in Martinec (2000): the field equations are solved in the spherical harmonic domain, but integrated over time with an explicit time differencing scheme. In view of solving the momentum equation in the spectral solution, the code is rather efficient and in view of the time-differencing scheme the coupling with the Earth-system
model is straight forward (Konrad et al., 2015). The loading is considered as a pressure-field boundary condition applied at the surface, where mass conservation is considered solving the sea level equation (Farrell and Clark, 1976) at each time step. The time step interval is constrained by the minimum ratio of viscosity vs. shear modulus (the Maxwell time). It is usually about $20\,\mathrm{yr}$ for a standard Earth structure but has to be reduced to about $1\,\mathrm{yr}$ for a 3D structure containing structural features like low viscous regions in tectonically active regions. For the 3D viscosity field in this study we chose the 3D model $v_1.0_s16$ from
Bagge et al. (2021) (Fig. D1) as it shows the smallest misfit with observational data in independent GIA models. In the present work we set a minimum viscosity of $10^{19.5}$, which allows a time step of $10\,\mathrm{yr}$ to be used. Due to the fact that the time step is rather small an iteration at each time step as discussed in Kendall et al. (2005) can be neglected.

The relative sea level determined by VILMA is used as the spatially variable correction of the considered reference topography, $h_{\mathrm{topo}}(t_0)$. In this way, changes in the topography are considered with respect to the changing geoid defined here as the
mean sea level at the respective time step:

$$h_{\mathrm{topo}}(t) = h_{\mathrm{topo}}(t_0) - h_{\mathrm{rsl}}(t - t_0)\,, \quad h_{\mathrm{rsl}}(t_0) = 0. \tag{D1}$$

This view is consistent with the natural definition of topograhy in the understanding of Carl Friedrich Gauss. It also means that all changes in elevation are expressed as measured relative to the mean sea level at that time.



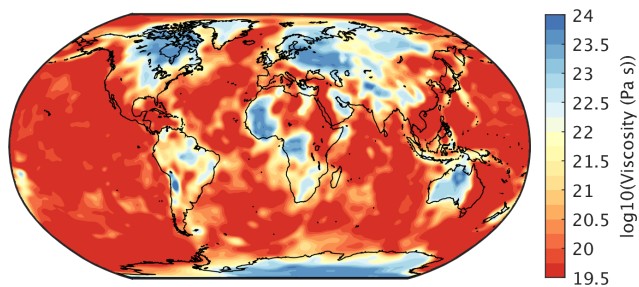

**Figure D1.** Viscosity at 200 km depth from Bagge et al. (2021) as used in the model simulations.

Changes in topography due to variations in sea level are considered furthermore by updating the land/sea mask at each time
step. This holds also for the conditions of floating vs. grounded ice. In view of completing formulations (e.g. Kendall et al.,
2005; Spada and Melini, 2019), the effect of rotational deformations was implemented recently (Klemann et al., 2023 in prep.)
following the formulation of Martinec and Hagedoorn (2014). Rotational deformations have to be considered, especially when
discussing the effect of GIA on geodetic observables like GNSS, EOP and gravity or when discussing future sea level changes
(e.g. Palmer et al., 2020).

*Author contributions.* MW, AG, RC and ST designed the study. RG provided the SICOPOLIS model code and RC, RG and JB assisted in the
implementation of SICOPOLIS into CLIMBER-X. RC and MW adapted and tested SICOPOLIS for the integration into CLIMBER-X. MW,
RC and AG developed SEMIX. VK and MB provided the VILMA code and contributed to its coupling. MW coupled, tested and tuned the
different model components. MW run the simulations and prepared the figures. MW wrote the paper, with contributions from all co-authors.

*Competing interests.* The authors declare that they have no conflict of interest.

*Acknowledgements.* M.W. and M.B. are funded by the German climate modeling project PalMod supported by the German Federal Ministry
of Education and Research (BMBF) as a Research for Sustainability initiative (FONA) (grant nos. 01LP1920B, 01LP1917D, 01LP1918A).
The authors gratefully acknowledge the European Regional Development Fund (ERDF), the German Federal Ministry of Education and
Research and the Land Brandenburg for supporting this project by providing resources on the high performance computer system at the
Potsdam Institute for Climate Impact Research.



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
