# Peer review of "Glacial inception through rapid ice area increase driven by albedo and vegetation feedbacks"

_EGUsphere, 2023_

## Author Response (AR1)

**Response to Reviewer #1**

We would like to thank the reviewer for the positive appraisal of our work and for the constructive comments on our paper. In blue below is our response to the reviewer comments and suggestions (in black italic). The line numbers in the response refer the revised version of the manuscript.

*I found this submission to be an interesting contribution to the field of coupled paleo-climate modeling. The study utilizes CLIMBER-X to effectively simulate the rapid ice growth over Eurasia and North America during this period and the ice decay after MIS 5d. The authors quantify the relative importance vegetation, ice sheet, and carbon cycle feedbacks play in the CLIMBER-X model for default parameters on ice growth and decay during the last glacial inception. They confirm the significant role dynamic vegetation plays in facilitating rapid ice growth, which is the most important of the tested feedbacks in their model simulations. Moreover, they confirm the importance of a temperature bias correction over North America for successful inception simulations with CLIMBER. However, their application of a summer bias correction throughout the year and constant through time must be explained. The bias correction enhances the agreement between simulated ice sheet configurations and geological records. The study's exploration of small temperature (+/- 1 degree C) and albedo (0.025) perturbations and their substantial influence on ice sheet volume and area adds further depth to the findings.*

*The content of this manuscript is relevant, shedding light on the complex dynamics of the last glacial inception and the factors influencing ice sheet growth and decay. The insights gained from this research with CLIMBER-X can contribute to the refinement of other paleo-climate models. However, the authors should comment on model and initialization uncertainty that can't be addressed using a single simulation/model realization per experiment.*

*I recommend this manuscript for publication with moderate edits. The authors have effectively addressed essential aspects of coupled paleo-climate modeling. With some adjustments, especially in explaining the assumptions made, this study will make a valuable addition to the body of literature in the field.*

**Major comments:**

*The introduction doesn't adequately prepare the reader for the results. What is actually novel in your study? While you present a literature review, you don't show clearly enough where this manuscript fits in, which previous issues it addresses and what new knowledge it will contribute. The only time the present work is addressed is in the last sentence ("In this study we employ the Earth System model CLIMBER-X (Willeit et al., 2022, 2023) with interactive ice sheets, viscoelastic solid Earth response and dynamic vegetation to simulate the last glacial inception from 125 ka to 100 ka.") which is not enough to guide the reader (who might not want to read the full paper but look for specific subsections) and stir interest. For example, lines 44-52 are unclear. Are you listing issues that previous studies had? Or important feedbacks that other studies have found that must be included to simulate the last*

*glacial inception successfully? Here would be a good time to mention how your work will include/improve/explore said feedbacks and findings*

We agree with the reviewer that the contribution of this study to our general understanding of the last glacial inception needs to be more clearly explained and emphasized in the introduction. We have therefore extended the introduction by a few paragraphs describing more clearly what has been achieved so far and what the advancements of the current work are in terms of understanding mechanisms and important processes at play during the glacial inception.

In the revised paper we also added an outline of the paper at the end of the introduction to guide the reader:

*"The paper first describes the model and in particular the ice sheet surface mass balance computation and the ice sheet coupling strategy (Section 2), with more details provided in the Appendix. Then the experimental setup is presented (Section 3) followed by a description of the results of the model simulations (Section 4) and finally discussions and conclusions."*

*Limitations of this non-ensemble approach, uncertainties in parametric values and model initialization should be clearly stated*

We assume that "ensemble" in this context means "perturbed physics ensemble" since in climate modelling, "simulation ensemble" or "ensemble of model simulations" have different meanings (see IPCC glossary). As far as the perturbed physics approach is concerned, we were among the first to apply it in paleoclimatology (e.g. Von Deimling et al. (2006)). While this approach has some merit, for example, for assessing uncertainties of future climate projections, it also has serious problems. The main problem is that in such an ensemble, only one member, the standard model version, is properly calibrated and tested; some members are less realistic, and some are completely unrealistic. In addition, it is known that a perturbed physics ensemble does not mimic a multi-model ensemble since the perturbation of model parameters does not reproduce the structural uncertainties. This is why, in most of our publications, we used another approach: we first carefully selected the baseline model version that had the most realistic performance and then performed a dedicated set of sensitivity experiments following the principle: one experiment, one model parameter change. We believe this approach is better suited for understanding the mechanisms of past climate variability.

We do not believe that there is a meaningful way to assess "uncertainties in parametric values", and this is why we never tried to do that. Instead, we study the sensitivity of model results to key model parameters.

Concerning model initialisation, there is no indication that climate was not in quasi-equilibrium at 125 ka. Therefore, except for carbon cycle processes, which are not relevant in this study because we use prescribed $CO_2$ concentrations, only the Greenland ice sheet would likely not be in quasi-equilibrium with 125 ka climate conditions. However, Greenland is clearly not the focus of our study and plays a negligible role in the glacial inception over the other NH continents.

We added this discussion to the revised manuscript (Lines 226-231):

*"The initial conditions of the model runs correspond to the climate model in equilibrium with 125 ka boundary conditions. We therefore make the reasonable assumption that climate was in quasi-equilibrium at 125 ka. Because the Greenland ice sheet was likely not in equilibrium with climate at this time, for practical reasons we choose to start from the Greenland ice sheet prescribed from present-day observations with a uniform ice temperature of -10°C.*

*This is justified because Greenland is not the focus of our study and plays a negligible role in the glacial inception over the other NH continents."*

*Application of the temperature bias correction (around line 118). Why would you apply the summer bias correction throughout the year? I don't see any reasonable explanation for that. What does the winter bias look like? And how can you assume the present-day bias is constant over time?*

The reasoning behind the choice of applying summer temperature bias correction throughout the year is guided by the fact that, following Milankovitch theory, ice sheet surface mass balance is largely determined by ablation during summer, which is highly sensitive to temperature. The winter temperature biases over North America are similar in pattern but larger than the summer biases (see Fig. 5 in Willeit et al. (2022)), but since temperatures are anyway below freezing during this time of the year, this bias has only a very limited effect on surface mass balance. We have actually also tested the use of bias correction applying monthly mean temperature anomalies, but the model results indicated absolutely negligible differences compared to using the mean summer bias. We therefore prefer the simpler and physically based choice of using the mean JJA temperature bias throughout the year. In the revised paper we will add some sentences explaining the reasoning behind the use of mean summer temperature bias correction.

Since the focus of our study is on the last glacial inception and the boundary conditions at ~120ka (just before the expected onset of ice sheet growth) are similar to pre-industrial in terms of GHGs concentrations and orbital parameters, we do not expect significant differences in temperature biases compared to the present-day. The assumption on stationarity of the temperature biases is therefore, at least during the initial ice growth phase, well justified. This obviously changes when substantial ice cover starts to develop over northern North America, but for this time period we have neither good paleoclimate data nor GCM model simulations available to test the assumption.

We added a few sentences discussing the assumption of stationarity of temperature biases along these lines (Lines 156-166):

*"The bias correction is applied throughout the year as a constant offset in the computation of the surface energy balance. The reasoning behind the choice of applying summer temperature bias correction throughout the year is guided by the fact that, following Milankovitch theory, ice sheet surface mass balance is largely determined by ablation during summer, which is highly sensitive to temperature. We have actually also tested the use of bias correction applying monthly mean temperature differences, but the model results indicated absolutely negligible differences compared to using the mean summer bias. We also assume that the bias is a persistent feature of the model also under different boundary conditions and therefore apply the same bias correction at all times. Since the focus of our study is on the last glacial inception and the boundary conditions at ~120 ka (just before the expected onset of ice sheet growth) are similar to pre-industrial in terms of GHGs concentrations and orbital parameters, we do not expect significant differences in temperature biases compared to the present-day. The assumption on stationarity of the temperature biases is therefore, at least during the initial ice growth phase, well justified."*

**Minor comments:**

*Line 23: Typo: Milanlkovitch*

Fixed, thanks.

*Line 29: "relatively well covered by paleoclimate data": is it well covered? What is "relatively"? Aren't there significant uncertainties in any pre-LGM geological reconstructions?*

Relative only to the previous glacial inceptions. The uncertainties of all pre-LGM reconstructions are large indeed. We clarified this in the revised manuscript:
*"Since the most recent glacial inception, which began between 120 and 115 ka, is, compared to previous glacial inceptions, relatively well covered by paleoclimate data…"*

*84: "while Antarctica is prescribed at its present-day state in this study": reasoning for this assumption?*

This is not an assumption, this a model setup since we wanted to concentrate only on the Northern Hemisphere. The assumption is that the AIS contribution to global ice volume during glacial inception is small. This is only relevant for comparison of simulated and reconstructed sea level. Most studies of glacial inception are limited to the Northern Hemisphere. In the publications where the Antarctic ice sheet was included, from the classical work by Huybrechts (2002) to the recent Albrecht et al. (2020), the Antarctic contribution to global sea level rise during MIS 5 is only about 5 msl, which is 10% of global sea level variations reconstructed for this period.
Following also the comment of Reviewer#2 we added the following text to provide further justification of the fixed-Antarctica setup (Lines 113-117):
*"Since we concentrate on the NH, Antarctica is prescribed at its present-day state in this study based on the assumption that the Antarctic ice sheet contribution to global ice volume changes during the glacial inception is small. Different modelling studies confirmed that during this time period the Antarctic contribution to global sea level drop is only about 5 m (Huybrechts (2002), Albrecht et al. (2020)), about 10% of global sea level variations reconstructed for this period."*
We also added a sentence discussing the potential role of Antarctic contribution to sea level change when discussing Fig. 4 (Lines 243-246):
*"There is an overall good agreement between simulated and reconstructed sea level in terms of timing, while the amplitude of the changes is somewhat underestimated (Fig. 4d). Part of this discrepancy could be related to the missing contribution from Antarctica, which is prescribed at its present-day state in our simulations and could have contributed ~5 m to the sea level drop (Huybrechts (2002), Albrecht et al. (2020))."*

*Line 99: "subsequently, temperature, humidity and radiation fields are downscaled onto the high-resolution topography.": can you account for orographically forced precipitation on the high-resolution topography?*

The term "downscaled" is misleading. Climatological fields are horizontally and vertically interpolated. We do not account for the additional orographic effect on precipitation. Such an effect is crudely accounted for by the atmospheric component of CLIMBER-X, which has a much higher resolution (5°x5°) compared to CLIMBER-2, where a parameterization of the orographically forced precipitation was included (Calov et al., 2005).
We replaced "*downscaled*" with "*elevation-corrected*" and added the following sentence (Lines 133-136):
*"We do not account for the additional orographic effect on precipitation. Such an effect is*

*crudely accounted for by the atmospheric component of CLIMBER-X, which has a much higher resolution compared to CLIMBER-2, where a parameterization of the orographically forced precipitation was included (Calov et al., 2005).*"

*Line 101: "concentration of dust in snow": what dust? Is there dust forcing? Are dust sources and transport simulated? Where the dust is coming from should be explained here in short and in more detail in the supplements.*

CLIMBER-X incorporates a fully interactive dust cycle, including atmospheric dust transport and dust deposition, as described in Willeit et al. (2022) and Fig. 11 and 12 in Willeit et al. (2023). Thus, the average concentration of dust in snow is computed by the model.
We added dust to the schematic in Fig. 1 and added the following sentence (Lines 138-139): "*CLIMBER-X includes an interactive dust cycle (Willeit et al. (2022) and Willeit et al. (2023)) and the dust concentration in snow is computed from the simulated dust deposition flux.*"

*Lines 117-118: "we implemented a temperature bias correction over northern North America that has a dipole structure": while explained in the supplements, it is not clear here if this is constructed or simply the JJA summer temperature field of ERA5 minus CLIMBER*

The temperature bias correction used in SEMIX is the difference in mean summer temperature between modern reanalysis data from ERA5 and CLIMBER-X near-surface air temperature fields. The "dipole structure" of this bias correction is mentioned mainly in relation to Ganopolski et al. (2010). Since we agree that it can be misleading, we removed the term 'dipole' when not explicitly referring to Ganopolski et al. (2010).

*Line 199: Cite/compare to snowfield glaciation versus spreading from high-elevation nucleation sites in Bahadory et al. 2021: Last glacial inception trajectories for the Northern Hemisphere from coupled ice and climate modelling*

We rewrote the paragraph, referring also to Bahadory 2021 (Lines 251-255):
"*At 121 ka the simulated ice sheet extent is comparable to the present-day, but with some loss of ice in southern Greenland (Fig 6a). Then ice starts to nucleate at high-altitudes in the Arctic Islands and over the mountain ranges of Scandinavia and the Ellsmere and Baffin Islands (Fig. 6b) before rapidly expanding over the Canadian Arctic Archipelago and Scandinavia, mainly due to large-scale thickening of snowfields (Fig. 6c). This is in line with what found by Bahadory et al. (2021). Starting from the Arctic islands, ice also spreads into the Barents and Kara Seas and the Cordilleran ice sheet is established (Fig. 6c).*"

*Table 1: unclear from the table what T offset, geo, and snow albedo offset are*

We expanded the caption of the table to include an explanation of the terms and abbreviations used.

*Figure 9 title: Zonal mean differences -> Northern hemisphere zonal mean differences*

Changed.

*Lines 232-236: structure: I'm missing a short experiment description before we dive into the results*

We have rewritten the paragraph to include a short experiment description (Lines 283-289):
*"The effect of the vegetation feedback on the expansion of ice sheets during glacial inception can be quantified by running a glacial inception simulation with vegetation prescribed at its pre-industrial state. Practically, fixed vegetation in our simulation means that the plant functional type fractions are not allowed to change and that the maximum leaf area index is fixed. However, for deciduous plants, the seasonality of the phenology will still be affected by the changing climatic conditions.*
*The vegetation feedback plays a crucial role for glacial inception in our simulations. This is clearly illustrated by the much smaller increase in ice sheet area and volume in simulations where vegetation is prescribed at its equilibrium pre-industrial state, compared to the reference glacial inception simulation with interactive vegetation (Fig. 12)."*

*Line 254: "higher albedo of ice compared to ice-free land": wouldn't most of the now ice-free land be snow-covered?*

This would not be the case during the ice-retreat phase, which is characterized by negative surface mass balance and snow-free conditions for at least part of the summer.

*Figure 14: why not also include the fixgeo experiment here?*

At 115ka, the Fixgeo experiment has a very similar ice sheet configuration compared to the reference run, so adding it to the figure would not provide much new information. The Fixgeo simulation differs substantially only during the ice retreat period.

*Figures 16, 17, 19: figure key consistently in the top panel like in other figures*

We have moved the figure legends to the top panel.

*Line 321: "This result is fully consistent with the concept of glacial inception as a bifurcation in the climate system": You haven't really introduced the concept, and I don't quite see how this plays a role here…*

We have now introduced the concept more prominently in the introduction and have additionally rewritten this sentence as (Lines 374-375):
*"This result is fully consistent with the concept of glacial inception as a bifurcation in the climate system caused by a strong albedo feedback (Calov et al., 2005)."*

*Line 334: "A climate acceleration factor of 10 would allow more complex Earth system models to run transient glacial inception simulations in a reasonable time using less computational resources.": Can we assume the finding still holds for more complex models? With the inclusion of more complex feedbacks and non-linearities, I would assume models can't be accelerated as much*

Complex Earth system models currently used for modelling glacial inception use large acceleration because, at present, it is not possible to run high-resolution models for 20,000 years. Our results provide some justification for such an approach.
We modified the sentence to make it clearer that it is not obvious that our finding is directly applicable also to GCMs (Lines 396-399):
*"The model results are not very sensitive to climate acceleration up to a factor ~10. Assuming that this finding also holds for more complex Earth system models, a climate*

*acceleration factor of 10 would allow these models to run transient glacial inception simulations in a reasonable time using less computational resources."*

*Line 401: "A constant temperature lapse rate is used": is assuming a constant lapse rate reasonable?*

This is a common assumption for such type of studies. We are not aware about any scientific basis for improving this approach.

*Line 453: missing equation reference*

Fixed, thanks.

*All ice sheet maps: the grayscale color key offers a poor discernable resolution*

We replaced the colour map with a more discrete one, which made the figures better readable.

*Potentially unnecessary figures if the paper needs to be shortened: Figure 3, 18*

We would like to keep Figure 3 as it shows that the model does not simulate glacial inception at present, which is an important constraint. We would also like to keep Figure 18 as it could provide useful information for glacial inception simulations with GCMs using climate acceleration techniques.

**References**

Albrecht, T., Winkelmann, R., and Levermann, A.: Glacial-cycle simulations of the Antarctic Ice Sheet with the Parallel Ice Sheet Model (PISM) – Part 2: Parameter ensemble analysis, Cryosph., 14, 633–656, https://doi.org/10.5194/tc-14-633-2020, 2020.

Bahadory, T., Tarasov, L., and Andres, H.: Last glacial inception trajectories for the Northern Hemisphere from coupled ice and climate modelling, Clim. Past, 17, 397–418, https://doi.org/10.5194/cp-17-397-2021, 2021.

Calov, R., Ganopolski, A., Claussen, M., Petoukhov, V., and Greve, R.: Transient simulation of the last glacial inception. Part I: glacial inception as a bifurcation in the climate system, Clim. Dyn., 24, 545–561, https://doi.org/10.1007/s00382-005-0007-6, 2005.

Von Deimling, T. S., Ganopolski, A., Held, H., and Rahmstorf, S.: How cold was the last Glacial maximum?, Geophys. Res. Lett., 33, 1–5, https://doi.org/10.1029/2006GL026484, 2006.

Ganopolski, A., Calov, R., and Claussen, M.: Simulation of the last glacial cycle with a coupled climate ice-sheet model of intermediate complexity, Clim. Past, 6, 229–244, https://doi.org/10.5194/cp-6-229-2010, 2010.

Huybrechts, P.: Sea-level changes at the LGM from ice-dynamic reconstructions of the Greenland and Antarctic ice sheets during the glacial cycles, Quat. Sci. Rev., 21, 203–231, https://doi.org/10.1016/S0277-3791(01)00082-8, 2002.

Willeit, M., Ganopolski, A., Robinson, A., and Edwards, N. R.: The Earth system model

CLIMBER-X v1.0 – Part 1: Climate model description and validation, Geosci. Model Dev., 15, 5905–5948, https://doi.org/10.5194/gmd-15-5905-2022, 2022.

Willeit, M., Ilyina, T., Liu, B., Heinze, C., Perrette, M., Heinemann, M., Dalmonech, D., Brovkin, V., Munhoven, G., Börker, J., Hartmann, J., Romero-Mujalli, G., and Ganopolski, A.: The Earth system model CLIMBER-X v1.0 – Part 2: The global carbon cycle, Geosci. Model Dev., 16, 3501–3534, https://doi.org/10.5194/gmd-16-3501-2023, 2023.

**Response to Reviewer #2**

We would like to thank the reviewer for the comments on our paper. In blue below is our response to the reviewer comments and suggestions (in black italic). The line numbers in the response refer the revised version of the manuscript.

*The manuscript presented by Willeit and co-workers is very much in line with other studies that have been conducted with similar complexity models at long time scales.*

We are only aware of a few (apart from our own) modelling attempts to simulate the last glacial inception with coupled climate-ice sheet models, and even fewer realistic ones.

*As such, the aim is fine and the research presented is sound. Adequate attention is given to model parametrisation and process analysis, which is the strong point of the presented research. As presented however, the manuscript is an hybrid between a development paper and a research paper. It clearly **lacks a research focus and is incremental on previous research with the same group**. The latter point is not an issue and should not prevent publication.*

We are grateful to the referee for not ruling out the possibility of publication of our article, but we respectfully disagree with the highlighted statement:

1. We think that the research question is clearly formulated: understanding the mechanism of glacial inception. In particular, why and how ice volume rises so rapidly immediately after the onset of glaciation. Based on other publications related to this topic, we believe that this is still a valid research question. Following also the comments from Reviewer#1 we added some more discussion in the introduction to make it clearer how our study contributes to the advancement in the understanding of the processes and feedbacks at play during glacial inception.

2. As we discuss below, for this study we use a new and superior Earth system model compared to CLIMBER-2, which we used for a similar study 20 years ago. It is quite common in climate modelling to revisit similar problems time after time as new and better tools are developed and become available. In the paleoclimate context, this is for example the case for numerous simulations of LGM climate performed during four rounds of PMIPs by more or less the same groups, with more or less similar models and with similar boundary conditions.

**Main concerns**

*1-/ Target journal and article format. The manuscript, as mentionned above is an hybrid between a development paper and a first research application paper. I find very surprising that the authors have chosen to pack all this in one manuscript, while other model developments manuscript are already published or in the way with the same first author in*

*Geoscientific Model Development. Personally, I recommend that the current manuscript is splitted in two parts : a model development manuscript that would cover the themes of ice-sheet coupling, sea-level prediction model coupling and snow mass balance computation (including comparison to present-day fields since this is crucial for ice-sheet evolution). This would remove the large appendixes in the current manuscript and allow proper discussion of the modeling choices made by the expert community. A second, much lighter manuscript, would target the inception question with the model in a Climate of the Past research artcile. I see no good reason to proceed the way that the author did, yielding a manuscript that is complicated to evaluate correctly since other discussion are missing in an already too long manuscript.*

CLIMBER-X is a very well documented Earth system model. There are already three published GMD papers describing different components of CLIMBER-X and their performance: Willeit and Ganopolski (2016), Willeit et al. (2022) and Willeit et al. (2023). In addition, individual components of CLIMBER-X, which were developed outside of PIK – GOLDSTEIN, HAMOCC, SICOPOLIS and VILMA - are described in separate papers. The purpose of a description paper is not only to present model equations but also to demonstrate model performance versus observational data and results of other models. Since in this study we are interested only in ice sheets of the Northern Hemisphere, the only ice sheet for which there are observational and good modelling data at present is Greenland. However, the performance of our model for Greenland has already been presented in Calov et al. (2018) (stand-alone SICOPOLIS model) and Höning et al. (2023) (CLIMBER-X in full configuration). We added these two references to the revised paper to make it clearer that the model evaluation for Greenland has already been performed elsewhere. For past climates, good modelling results exist only for the LGM, but we have already discussed the CLIMBER-X performance for the LGM in Willeit et al. (2022). Moreover, a more detailed comparison of ice sheet surface mass balance with e.g. PMIP models for the LGM would be of very limited use, as it was shown that the different GCMs produce very different LGM climates that would lead to simulated ice sheets varying widely among models (Niu et al., 2019). Thus, it is unclear what else we could present in a fourth CLIMBER-X description paper proposed by the Referee, except for the set of equations shown in the Appendix.

*Two examples of this issue :*

*line 114-120, page 5. Discussion of the constant bias correction over time in SEMIX is one example where this should be much more discussed and potentially be evaluated : we have many simulations in the different PMIP phases with GCMs that are performed for interglacials where you could test the validity of your stationarity of the bias (using other model anomalies and your own).*

We do not understand how simulations of other interglacials can be used to test the stationarity of the temperature bias. The bias is the difference between model simulations and observational (reanalysis) data. Obviously, there are no real observations from the past and paleoclimate reconstructions are extremely uncertain. Thus, model biases for paleoclimates cannot be evaluated in principle. GCMs have their own climate biases, which are comparable in magnitude with the biases of CLIMBER-X, and can therefore not be used as a reference. However, since the focus of our study is on the last glacial inception and the boundary conditions at ~120ka (just before the expected onset of ice sheet growth) are similar to pre-industrial in terms of GHGs concentrations and orbital parameters, we do not expect significant differences in temperature biases compared to the present-day. The assumption on

stationarity of the temperature biases is therefore, at least during the initial ice growth phase, well justified. This obviously changes when substantial ice cover starts to develop over northern North America, but for this time period we have neither paleoclimate data nor GCM model simulations available to test the assumption.

We added a few sentences discussing the assumption of stationarity of temperature biases along these lines (Lines 161-166):

*"We also assume that the bias is a persistent feature of the model also under different boundary conditions and therefore apply the same bias correction at all times. Since the focus of our study is on the last glacial inception and the boundary conditions at ~120 ka (just before the expected onset of ice sheet growth) are similar to pre-industrial in terms of GHGs concentrations and orbital parameters, we do not expect significant differences in temperature biases compared to the present-day. The assumption on stationarity of the temperature biases is therefore, at least during the initial ice growth phase, well justified."*

*Line 100 : « fields are downscaled onto the high-resolution topography ». This is totally insufficient since most of the results of this manuscript and all the forthcoming with CLIMBER-X are dependent on the details of this downscaling. A detailed evaluation of the downscaling for rough topography should be given in a development manuscript. The current description of the equations in the appendix B1 is clear, but the absence of evaluation is unacceptable.*

The term "downscaled" is misleading in this context. As described in Appendix B1, we only use horizontal and vertical interpolation of climatic fields and use a standard temperature correction for elevation with a constant lapse rate. We replaced the term 'downscaled' with 'elevation-corrected' to make it clearer that we are not performing any dynamic downscaling, but a simple bilinear interpolation with elevation correction.

*2-/ The main message of the manuscript is the relationship between surface albedo change (snow cover extent) and vegetation feedbacks that are at the source of the rapid ice-sheet expansion simulated, not the ice-volume. This is a fine conclusion, but is also one that has been largely promoted already by the same group (Ganopolski et al., 2010, doi :10.5194/cp-6-229-2010) with the previous generation of their model. In the current version of the manuscript, little is said about the comparison to this previous results.*

Actually, not in Ganopolski et al. (2010) but in Calov et al. (2005a) and Calov et al. (2005b), i.e. nearly 20 years ago. Since then, this conclusion has neither been confirmed nor disproved. The new paper revisits this problem with a new modelling tool. The new results confirm our earlier finding that a rapid expansion of the ice area at the beginning of glacial inception is crucial for simulating realistic ice volume during MIS5d. We now discuss this in some detail in the introduction and in the conclusion sections. The importance of this finding crucially depends on the degree of novelty of our modelling tool. The referee wrote:

***Given the amount of components that are shared with the CLIMBER-2 model**, it is in my view a requirement that such a comparison is made to assess what is really new in the study presented or at least what is mostly the same model response and what is not.*

CLIMBER-X is a new Earth system model which does not share any modelling components with CLIMBER-2. The only things they have in common are the acronym and the fact that both these models (as well as CLIMBER-3) were developed in the group led by the last author of the manuscript. Although CLIMBER-X does employ a simplified (compared to

modern AGCMs) atmospheric model, it is clear from Willeit et al. (2022), that this model, SESAM, is much more advanced than the atmospheric model in CLIMBER-2 (Petoukhov et al., 2000); the ocean in CLIMBER-2 is a zonally averaged model (Stocker et al., 1992) while in CLIMBER-X we use the 3-D ocean model GOLDSTEIN (Edwards et al., 1998; Edwards and Marsh, 2005; Edwards and Shepherd, 2002). Similarly, the sea ice component in CLIMBER-2 is 1-D and in CLIMBER-X it is 2D with an explicit treatment of sea ice transport; the integrated land-vegetation model PALADYN (Willeit and Ganopolski, 2016) employed in CLIMBER-X is by complexity and degree of realism absolutely incomparable with the simplistic land and vegetation components of CLIMBER-2 (Brovkin et al., 1997; Petoukhov et al., 2000). The situation is similar with the ice sheet component, SICOPOLIS. Ralf Greve used the same generic name SICOPOLIS for the ice sheet models which he developed during the past ~30 years. The SICOPOLIS version that was used in CLIMBER-2 is a shallow ice approximation model typical for the 1990s, with the simplest local bedrock relaxation parameterization. SICOPOLIS v5.3 (https://doi.org/10.5281/zenodo.6872648), which is used in CLIMBER-X, is a modern hybrid model with a proper treatment of ice streams and ice shelves. Furthermore, CLIMBER-X is coupled with the state-of-the-art 3D solid Earth model VILMA (Klemann et al., 2008; Martinec et al., 2018). Last but not least, CLIMBER-2 has a climate component resolution of 51°x10° and an ice sheet resolution of about 100 km, while CLIMBER-X has a horizontal resolution of 5°x5° and (in this study) 32 km, respectively. In terms of grid cells by unit of area this means an increase by an order of magnitude for both climate and ice sheet components. Higher spatial resolution not only improves model performance, but also allows us to introduce time-dependent land/sea mask and river routing as well as evolution of proglacial lakes, features which for obvious reasons were absent in CLIMBER-2. Thus, CLIMBER-X is new and superior in all respects compared to the 25 years old CLIMBER-2. This is why we think that applying this model to scientific problems which we already studied with CLIMBER-2 is fully justified.

To make it clearer to the reader that CLIMBER-2 and CLIMBER-X are fundamentally different models, we have added some more detailed discussion of the CLIMBER-X model in the introduction and in the methods, in addition to the references to the GMD papers describing CLIMBER-X.

*3-/ The discussion of the Figure 7 is not at an appropriate scientific level. Line 208 mentions that the model and data « compares reasonably well » which falls short of the mark. There is then a few consideration on the different places where the model is glaciated and not. However, there is in my view a fundamental problem in the ice evolution presented at 117 and 115 ka. Cited reconstructions seem to indicate that there is a double semi-independent ice-sheet build up, on one side on the Cordilleran ice-sheet and second over the northern part of Canada with more expansion over land in the Nunavut and Quebec areas. The model to the contrary indicates a very zonal expansion over the Hudson Bay, which is not what is indicated in the reconstructions (that are uncertain, but clearly indicate this more extensive expansion over Canada). The model also have a tendency to merge the two ice-sheets (Cordilleran and Nunavut/Northern Territories). What is the impact of all this on the results ?*

As acknowledged by the reviewer, the general introductory sentence "compares reasonably well…" is followed by a whole paragraph discussing agreement and disagreement between model and reconstructions over the different regions. We believe that a more quantitative ('scientific'?) comparison of model and reconstructions is clearly not possible, given the poor quality of the ice sheet reconstructions. Any pre-LGM ice sheet reconstruction should be treated as a rough representation of sparse empirical data supported by some modelling

exercises. This is for instance the case for the Batchelor et al. (2019) reconstruction, where the table in the Supplementary Information describes the sources of information used to reconstruct ice sheets at MIS 5d. For North America they are: (1) single empirical outline (sketch) from (Kleman et al., 2010), the dating of which is uncertain (5b or 5d); (2) results of several model simulations, including our own simulations with CLIMBER-2 (Ganopolski and Calov, 2011) and another CLIMBER-2 simulation performed at LSCE (Bonelli et al., 2009); (3) a single empirical data point which is used to constrain the ice sheet extent in the eastern part of North America (Figure 6c Supplementary Information, Batchelor et al. 2019). This large uncertainty is also reflected in the "min", "best" and "max" reconstructions in Batchelor et al. (2019). We performed a set of quasi-equilibrium experiments using CLIMBER-X constrained by the ice extent of all Batchelor's reconstructions (S. Talento pers. communication) and found that the "best" 5d reconstruction significantly underestimates global ice sheet volume at that time while "max" significantly overestimates it. Whether one can conclude from this fact that the truth is somewhere in-between of "best" and "max" – we do not know.

*Likewise, very little is said about the clear expansion of the ice-sheet in Alaska (obvious tendency in figure 14 that is only partially corrected) and which was already a persistent feature of ice-sheet simulations in CLIMBER-2.*

'Excessive' glaciation of Alaska at the LGM isn't a persistent feature only of CLIMBER-2 but also of other models of the same class and more complex models. As far as the glaciation of western North America during MIS 5d is concerned, it is not constrained by paleodata, and one can only guess how much ice was in Alaska during this time. Reconstructions seem to indicate that there was some ice over Eastern Siberia during MIS 5d but not at LGM and it is therefore reasonable to think that Alaska was closer to glaciation during the last inception than at the LGM.

*4-/ There is no discussion of the potential impact of a fixed Antarctic ice-sheet. It is mentionned at the beginning but then totally ignored. This is very much worrying since the authors are simulating sea-level. At the very least, a discussion of the potential impact, limitations etc. should be included.*

We agree that such discussion is needed. However, it is unclear why the Referee considers not modelling the Antarctic ice sheet to be "very much worrying". Most studies of glacial inception are limited to the Northern Hemisphere. The assumption is that the AIS contribution to global ice volume during glacial inception is small and it is relevant mainly for comparison of simulated and reconstructed sea level. In the publications where the Antarctic ice sheet was included, from the classical Huybrechts (2002) and to the recent Albrecht et al. (2020), the Antarctic contribution to global sea level rise during MIS 5 is only about 5 msl, which is 10% of global sea level variations reconstructed for this period. Following also the comment of Reviewer#1 we added the following text to provide further justification of the fixed-Antarctica setup (Lines 113-117):
*"Since we concentrate on the NH, Antarctica is prescribed at its present-day state in this study based on the assumption that the Antarctic ice sheet contribution to global ice volume changes during the glacial inception is small. Different modelling studies confirmed that during this time period the Antarctic contribution to global sea level drop is only about 5 m (Huybrechts (2002), Albrecht et al. (2020)), about 10% of global sea level variations reconstructed for this period."*
We also added a sentence discussing the potential role of Antarctic contribution to sea level

change when discussing Fig. 4 (Lines 243-246):

*"There is an overall good agreement between simulated and reconstructed sea level in terms of timing, while the amplitude of the changes is somewhat underestimated (Fig. 4d). Part of this discrepancy could be related to the missing contribution from Antarctica, which is prescribed at its present-day state in our simulations and could have contributed ~5 m to the sea level drop (Huybrechts (2002), Albrecht et al. (2020))."*

*5-/ In many places in the manuscript, the term « carbon cycle feedback » is used, but for me there is no carbon cycle feedback simulation in this research. The pCO2 of the atmosphere is fixed to reconstructions and vegetation is simulated on land (so probably the carbon as well) but not discussed. If the authors means « vegetation feedbacks » which is my guess there, then it should be corrected accordingly.*

The Referee is right. In the case of the interactive carbon cycle, as it was in the studies by Ganopolski and Brovkin (2017) and Willeit et al. (2019), the climate-carbon cycle feedback was explicitly modelled and discussed. In the current study, since CO2 concentration is prescribed, the term "the effect of CO2" is more appropriate than "carbon cycle feedback" and has be changed accordingly.

**Minor concerns**

*1-/ line 22, page 2 « interglacial (no significant ice sheets over the northern continents) ». Please reformulate. I have a hard time not finding the current Greenland ice-sheet not significant.*

We reformulated this to account for Greenland:
*"(no significant ice sheets over the northern continents, except for Greenland)"*

*2-/ line 236, page 14, « deciduos » → « deciduous »*

Fixed, thanks.

*3-/ line 172 : « continuosly » → « continuously »*

Fixed, thanks.

*4-/ line 181-182 : justification of such a starting point for the ice-sheet model is not justified. Why equilibrium at 125ka and not another condition ? How is this impacting the dynamics of the first part of your experiment, not having an transient evolution at the start ?*

There is no indication that climate was not in quasi-equilibrium at 125 ka. Therefore, except for carbon cycle processes, which are not relevant in this study because we use prescribed CO2 concentrations, only the Greenland ice sheet would likely not be in quasi-equilibrium with 125 ka climate conditions. However, Greenland is clearly not the focus of our study and

plays only a very limited role in the glacial inception over the other NH continents.
We added this discussion to the revised manuscript (Lines 226-231):
*"The initial conditions of the model runs correspond to the climate model in equilibrium with 125 ka boundary conditions. We therefore make the reasonable assumption that climate was in quasi-equilibrium at 125 ka. Because the Greenland ice sheet was likely not in equilibrium with climate at this time, for practical reasons we choose to start from the Greenland ice sheet prescribed from present-day observations with a uniform ice temperature of -10°C. This is justified because Greenland is not the focus of our study and plays a negligible role in the glacial inception over the other NH continents."*

*5-/ line 184 : another instance of « carbon cycle feedback », misleading*

Agreed: we changed "carbon cycle feedback" to "the role of CO2 variations".

**References**

Albrecht, T., Winkelmann, R., and Levermann, A.: Glacial-cycle simulations of the Antarctic Ice Sheet with the Parallel Ice Sheet Model (PISM) – Part 2: Parameter ensemble analysis, Cryosph., 14, 633–656, https://doi.org/10.5194/tc-14-633-2020, 2020.

Batchelor, C. L., Margold, M., Krapp, M., Murton, D. K., Dalton, A. S., Gibbard, P. L., Stokes, C. R., Murton, J. B., and Manica, A.: The configuration of Northern Hemisphere ice sheets through the Quaternary, Nat. Commun., 10, 1–10, https://doi.org/10.1038/s41467-019-11601-2, 2019.

Bonelli, S., Charbit, S., Kageyama, M., Woillez, M.-N., Ramstein, G., Dumas, C., and Quiquet, a.: Investigating the evolution of major Northern Hemisphere ice sheets during the last glacial-interglacial cycle, Clim. Past Discuss., 5, 1013–1053, https://doi.org/10.5194/cpd-5-1013-2009, 2009.

Brovkin, V., Ganopolski, A., and Svirezhev, Y.: A continuous climate-vegetation classification for use in climate-biosphere studies, Ecol. Modell., 101, 251–261, 1997.

Calov, R., Ganopolski, A., Claussen, M., Petoukhov, V., and Greve, R.: Transient simulation of the last glacial inception. Part I: glacial inception as a bifurcation in the climate system, Clim. Dyn., 24, 545–561, https://doi.org/10.1007/s00382-005-0007-6, 2005a.

Calov, R., Ganopolski, A., Petoukhov, V., Claussen, M., Brovkin, V., and Kubatzki, C.: Transient simulation of the last glacial inception. Part II: Sensitivity and feedback analysis, Clim. Dyn., 24, 563–576, https://doi.org/10.1007/s00382-005-0008-5, 2005b.

Calov, R., Beyer, S., Greve, R., Beckmann, J., Willeit, M., Kleiner, T., Rückamp, M., Humbert, A., and Ganopolski, A.: Simulation of the future sea level contribution of Greenland with a new glacial system model, Cryosphere, 12, 3097–3121, https://doi.org/10.5194/tc-12-3097-2018, 2018.

Edwards, N. and Shepherd, J.: Bifurcations of the thermohaline circulation in a simplified three-dimensional model of the world ocean and the effects of inter-basin connectivity, Clim. Dyn., 19, 31–42, https://doi.org/10.1007/s00382-001-0207-7, 2002.

Edwards, N. R. and Marsh, R.: Uncertainties due to transport-parameter sensitivity in an efficient 3-D ocean-climate model, Clim. Dyn., 24, 415–433, https://doi.org/10.1007/s00382-004-0508-8, 2005.

Edwards, N. R., Willmott, A. J., and Killworth, P. D.: On the Role of Topography and Wind Stress on the Stability of the Thermohaline Circulation, J. Phys. Oceanogr., 28, 756–778, https://doi.org/10.1175/1520-0485(1998)028<0756:OTROTA>2.0.CO;2, 1998.

Ganopolski, A. and Brovkin, V.: Simulation of climate, ice sheets and CO2 evolution during the last four glacial cycles with an Earth system model of intermediate complexity, Clim. Past Discuss., 1–38, https://doi.org/10.5194/cp-2017-55, 2017.

Ganopolski, A. and Calov, R.: The role of orbital forcing, carbon dioxide and regolith in 100 kyr glacial cycles, Clim. Past, 7, 1415–1425, https://doi.org/10.5194/cp-7-1415-2011, 2011.

Ganopolski, A., Calov, R., and Claussen, M.: Simulation of the last glacial cycle with a coupled climate ice-sheet model of intermediate complexity, Clim. Past, 6, 229–244, https://doi.org/10.5194/cp-6-229-2010, 2010.

Höning, D., Willeit, M., Calov, R., Klemann, V., Bagge, M., and Ganopolski, A.: Multistability and Transient Response of the Greenland Ice Sheet to Anthropogenic CO 2 Emissions, Geophys. Res. Lett., 50, 1–11, https://doi.org/10.1029/2022GL101827, 2023.

Huybrechts, P.: Sea-level changes at the LGM from ice-dynamic reconstructions of the Greenland and Antarctic ice sheets during the glacial cycles, Quat. Sci. Rev., 21, 203–231, https://doi.org/10.1016/S0277-3791(01)00082-8, 2002.

Klemann, V., Martinec, Z., and Ivins, E. R.: Glacial isostasy and plate motion, J. Geodyn., 46, 95–103, https://doi.org/10.1016/j.jog.2008.04.005, 2008.

Martinec, Z., Klemann, V., van der Wal, W., Riva, R. E. M., Spada, G., Sun, Y., Melini, D., Kachuck, S. B., Barletta, V., Simon, K., A, G., and James, T. S.: A benchmark study of numerical implementations of the sea level equation in GIA modelling, Geophys. J. Int., 215, 389–414, https://doi.org/10.1093/gji/ggy280, 2018.

Niu, L., Lohmann, G., Hinck, S., Gowan, E. J., and Krebs-Kanzow, U.: The sensitivity of Northern Hemisphere ice sheets to atmospheric forcing during the last glacial cycle using PMIP3 models, J. Glaciol., 65, 645–661, https://doi.org/10.1017/jog.2019.42, 2019.

Petoukhov, V., Ganopolski, a., Brovkin, V., Claussen, M., Eliseev, a., Kubatzki, C., and Rahmstorf, S.: CLIMBER-2: a climate system model of intermediate complexity. Part I: model description and performance for present climate, Clim. Dyn., 16, 1–17, https://doi.org/10.1007/PL00007919, 2000.

Stocker, T. F., Mysak, L. A., and Wright, D. G.: A Zonally Averaged, Coupled Ocean-Atmosphere Model for Paleoclimate Studies, J. Clim., 5, 773–797, https://doi.org/10.1175/1520-0442(1992)005<0773:AZACOA>2.0.CO;2, 1992.

Willeit, M. and Ganopolski, A.: PALADYN v1.0, a comprehensive land surface–vegetation–carbon cycle model of intermediate complexity, Geosci. Model Dev., 9, 3817–3857, https://doi.org/10.5194/gmd-9-3817-2016, 2016.

Willeit, M., Ganopolski, A., Calov, R., and Brovkin, V.: Mid-Pleistocene transition in glacial cycles explained by declining CO 2 and regolith removal, Sci. Adv., 5, eaav7337, https://doi.org/10.1126/sciadv.aav7337, 2019.

Willeit, M., Ganopolski, A., Robinson, A., and Edwards, N. R.: The Earth system model CLIMBER-X v1.0 – Part 1: Climate model description and validation, Geosci. Model Dev., 15, 5905–5948, https://doi.org/10.5194/gmd-15-5905-2022, 2022.

Willeit, M., Ilyina, T., Liu, B., Heinze, C., Perrette, M., Heinemann, M., Dalmonech, D., Brovkin, V., Munhoven, G., Börker, J., Hartmann, J., Romero-Mujalli, G., and Ganopolski, A.: The Earth system model CLIMBER-X v1.0 – Part 2: The global carbon cycle, Geosci.

Model Dev., 16, 3501–3534, https://doi.org/10.5194/gmd-16-3501-2023, 2023.